# Random Projections with Asymmetric Quantization

**Xiaoyun Li**
Department of Statistics
Rutgers University
Piscataway, NJ 08854
xiaoyun.li@rutgers.edu

**Ping Li**
Cognitive Computing Lab
Baidu Research USA
Bellevue, WA 98004
liping11@baidu.com

## Abstract

The method of random projection has been a popular tool for data compression, similarity search, and machine learning. In many practical scenarios, applying quantization on randomly projected data could be very helpful to further reduce storage cost and facilitate more efficient retrievals, while only suffering from little loss in accuracy. In real-world applications, however, data collected from different sources may be quantized under different schemes, which calls for a need to study the asymmetric quantization problem. In this paper, we investigate the cosine similarity estimators derived in such setting under the Lloyd-Max (LM) quantization scheme. We thoroughly analyze the biases and variances of a series of estimators including the basic simple estimators, their normalized versions, and their debiased versions. Furthermore, by studying the monotonicity, we show that the expectation of proposed estimators increases with the true cosine similarity, on a broader family of stair-shaped quantizers. Experiments on nearest neighbor search justify the theory and illustrate the effectiveness of our proposed estimators.

## 1 Introduction

The method of random projections (RP) [35] has become a popular technique to reduce data dimensionality while preserving distances between data points, as guaranteed by the celebrated Johnson-Lindenstrauss (J-L) Lemma and variants [24, 12, 1]. Given a high dimensional dataset, the algorithm projects each data point onto a lower-dimensional random subspace. There is a very rich literature of research on the theory and applications of random projections, such as clustering, classification, near neighbor search, bio-informatics, compressed sensing, etc. [22, 10, 4, 6, 8, 17, 18, 28, 15, 7, 19, 11, 9].

In recent years, "random projections + quantization" has been an active research topic. That is, the projected data, which are in general real-valued (i.e., infinite precision), are quantized into integers in a small number of bits. Applying quantization on top of random projections has at least two major advantages: (i) the storage cost is further (substantially) reduced; and (ii) some important applications such as hashing-table-based near neighbor search, require using quantized data for indexing purposes.

The pioneering example of quantized random projections should be the so-called "1-bit" (sign) random projections, initially used for analyzing the MaxCut problem [20] and then was adopted for near neighbor search [8] and compressed sensing [5, 23, 25]. As one would expect, storing merely 1-bit per projected data value in many situations might suffer from a substantial loss of accuracy, compared to using random projections with full (infinite) precision. There have been various studies on (symmetrically) quantized random projections beyond the 1-bit scheme, e.g., [13, 37, 26, 29, 31]. In this paper, we further move to studying "asymmetric quantization" of random projections, a relatively new problem arising from practical scenarios which is also mathematically very interesting.

Everyday, the process of data collection is taking place in every possible place that one can think of, but it is often impractical to cast a universal encoding strategy on data storage methods for every place. As a consequence, it becomes a meaningful task to look into the estimation problems with data encoded by different algorithms, or namely, the asymmetric case. In this paper, we provide

some insights on this type of problems, and particularly, we consider recovering inner products from asymmetrically quantized random projections, arising from the following two practical scenarios.

- **Scenario 1: quantization vs. full-precision.** Consider, for example, a retrieval system which uses random projections to process every data vector. To save storage, the projected data stored in the repository are quantized into a small number of bits. When a query data vector arrives, it is first processed by random projections. We then have the option of quantizing the projected query data vector before conducting the similarity search (with vectors in the repository); but we do not have to do the quantization step since we still have the projected query data vector in full-precision (why waste?). This situation hence creates the "quantization vs. full-precision" estimation problem. This setting is natural and practical, and the estimation problem has been studied in the literature, for example [14, 21, 27].

- **Scenario 2: quantization with different bits.** In applications such as large ad hoc networks [36, 30], data are collected and processed by different nodes (e.g., sensors or mobile devices) at different locations before sent to the central unit or cloud server. However, distinct nodes may use different quantization methods (or different bits) due to many possible reasons, e.g., memory capacity or purpose of data usage. In this situation, information retrieval among data sources using different quantization schemes could be on the cards. As a tightly related topic, asymmetric distributed source coding (with different bits from different sources) has also been considered in [3, 34] among others for sensor networks.

Scenario 1 is in fact an important special case of Scenario 2, where one source of data is quantized with infinite bits. In this paper, we provide thorough statistical analysis on the above two scenarios.

**Our contributions.** The major contributions of this paper include the following:

- In Section 3, we provide the bias and variance of linear and normalized inner product estimators in Scenario 1. We reveal an interesting connection between the variance of debiased inner product estimator and similarity search, which is very helpful in practice.

- In Sections 4 and 5, we conduct statistical analysis in Scenario 2, and prove the monotonicity of a large family of asymmetric quantized inner product estimators, which assures their validity for practical use. A new bound on the bias is also derived in the symmetric case.

- In Section 6, an empirical study on various real-world datasets confirms the theoretical findings and well illustrates the effectiveness of proposed quantization schemes.

## 2 Preliminaries

**Random Projections.** Let $U = [u_1, ..., u_n]^\mathrm{T} \in \mathbb{R}^{n \times d}$ be the original data matrix (with $d$ possibly being large). Random projections are realized by $Z = [z_1, ..., z_n]^\mathrm{T} = U \times R$, where $R \in \mathbb{R}^{d \times k}$, $k \ll d$ is a random matrix with i.i.d. standard Gaussian entries. Let $\|\cdot\|_2$ denote the $l_2$ Euclidean norm. Throughout this paper, we assume that every data point is normalized to unit norm[1], i.e., $\|u_i\|_2 = 1$, $1 \leq i \leq n$. We will hence use the terms "inner product" and "cosine similarity" interchangeably.

For the convenience of presentation, our results (estimators and properties) will be given for two pairs of data vectors, $u_i$ and $u_j$ (and correspondingly $z_i$ and $z_j$). Let $\rho = \langle u_i, u_j \rangle$ be the inner product between $u_i$ and $u_j$. We also denote $x = z_i$ and $y = z_j$. It is then easy to verify that $(x, y)$ is bi-variate normal:

$$\begin{pmatrix} x \\ y \end{pmatrix} \sim N\left( \begin{pmatrix} 0 \\ 0 \end{pmatrix}, \begin{pmatrix} 1 & \rho \\ \rho & 1 \end{pmatrix} \right). \tag{1}$$

**Lloyd-Max (LM) quantization [33, 32].** Assume a random signal model with signals generated from a probability distribution with density $X \sim f$. An $M$-level scalar quantizer $q_M(\cdot)$ is specified by $M + 1$ decision borders $t_0 < t_1 < \cdots < t_M$ and $M$ reconstruction levels (or codes) $\mu_i$, $i = 1, ..., M$, with the quantizing operator defined as

$$q_M(x) = \mu_{i^*}, \ i^* = \{i : t_{i-1} < x \leq t_i, \ 1 \leq i \leq M\}. \tag{2}$$

The "distortion" is an important quantity that measures how much information is lost from the original signal due to quantization. In this paper, we will also assume $M = 2^b$, with $b = 1, 2, ...$, being the number of bits used for the quantizer. Thus, we will write $q_b(\cdot)$ instead of $q_M(\cdot)$.

**Definition 1.** *The distortion of a b-bit quantizer $Q_b(\cdot)$ with respect to distribution $f$ is defined as*

$$\mathbb{E}\left((X - q_b(X))^2\right) = \int (x - q_b(x))^2 f(x) dx = \sum_{i=1}^{2^b} \int_{t_{i-1}}^{t_i} (x - \mu_i)^2 f(x) dx. \tag{3}$$

In this paper, $f$ is the standard normal, i.e., $f(x) = \phi(x) = \frac{1}{\sqrt{2\pi}} e^{-x^2/2}$ in the conventional notation for Gaussian. Also, we will use $Q_b$ to denote Lloyd-Max (LM) quantizer which minimizes the distortion and $D_b$ to denote the corresponding value of distortion:

$$Q_b = \underset{q}{\mathrm{argmin}}\, \mathbb{E}\left((X - q_b(X))^2\right), \quad D_b = \mathbb{E}\left((X - Q_b(X))^2\right) \tag{4}$$

A basic identity of LM quantizer is that $\mathbb{E}(Q_b^2(X)) = \mathbb{E}(Q_b(X)X)$. In practice, Lloyd's algorithm [32] is used to find the solution, which alternates between updating borders and reconstruction points until convergence (and the convergence is guaranteed).

**Estimates using full-precision RP's.** Consider observations $\begin{pmatrix} x_i \\ y_i \end{pmatrix} \overset{\text{i.i.d.}}{\sim} N\left(\begin{pmatrix} 0 \\ 0 \end{pmatrix}, \begin{pmatrix} 1 & \rho \\ \rho & 1 \end{pmatrix}\right)$, $1 \leq i \leq k$, as in (1). The task is to estimate $\rho$. One can use the usual simple estimator

$$\hat{\rho}_f = \frac{1}{k} \sum_{i=1}^{k} x_i y_i, \quad \text{with } \mathbb{E}(\hat{\rho}_f) = \rho, \quad Var(\hat{\rho}_f) = \frac{1 + \rho^2}{k}. \tag{5}$$

where $\mathbb{E}(\hat{\rho})$ is the expectation and $Var(\hat{\rho})$ is the variance. Note that the variance grows as $|\rho|$ increases. One can take advantage of the following so-called "normalized estimator":

$$\hat{\rho}_{f,n} = \frac{\sum_{i=1}^{k} x_i y_i}{\sqrt{\sum_{i=1}^{k} x_i^2} \sqrt{\sum_{i=1}^{k} y_i^2}}, \quad \mathbb{E}(\hat{\rho}_{f,n}) = \rho + O(\frac{1}{k}), \quad Var(\hat{\rho}_{f,n}) = \frac{(1 - \rho^2)^2}{k} + O(\frac{1}{k^2}). \tag{6}$$

$\hat{\rho}_{f,n}$ is nearly unbiased but it substantially reduces the variance especially near two extreme points $\rho = \pm 1$. We refer readers to the classical textbook [2] and recent papers [28, 27] for more details.

**Estimates using symmetric LM quantized RP's.** [29] study the inner product estimator under LM quantization scheme, by analyzing the biases and variances of estimators in the symmetric case. That is, the observations $x_i$ and $y_i$ are quantized by the same LM scheme with the same number of bits ($b$). In this paper, we study the asymmetric setting by using $b_1$ number of bits for quantizing $x_i$ and $b_2$ number of bits for $y_i$. Apparently, the work of [29] is a special case of our results (i.e., $b_1 = b_2$). Interestingly, our analysis also leads to a more refined bound on the estimation bias in the symmetric case compared to the corresponding bound in [29]. See Section 4 for the detailed results.

## 3 Scenario 1: Quantization vs. Full-precision

Recall that, we have i.i.d. observations $\{x_i, y_i\}$, $i = 1, 2, ..., k$, from a standard bi-variate normal with $x_i \sim N(0,1)$, $y_i \sim N(0,1)$, and $\mathbb{E}(x_i y_i) = \rho$. In this section, we study Scenario 1: quantization vs. full-precision. That is, we quantize $x_i$ with $b$ bits and we leave $y_i$ intact. The task is to estimate $\rho$ from $\{Q_b(x_i), y_i\}$, $i = 1, 2, ..., k$. We can still try to use the simple estimator similar to (5):

$$\hat{\rho}_{b,f} = \frac{1}{k} \sum_{i=1}^{k} Q_b(x_i) y_i. \tag{7}$$

As one would expect, this estimator $\hat{\rho}_{b,f}$ is no longer unbiased. We can show that $\mathbb{E}(\hat{\rho}_{b,f}) = \xi_{1,1} \rho$. Hence, we can attempt to remove the bias by using the following "debiased estimator"

$$\hat{\rho}_{b,f}^{db} = \frac{\hat{\rho}_{b,f}}{\xi_{1,1}} = \frac{1}{k} \frac{1}{\xi_{1,1}} \sum_{i=1}^{k} Q_b(x_i) y_i. \tag{8}$$

We will need to define $\xi_{1,1}$. More generally and analogous to the notation in [29], we define

$$\gamma_{\alpha,\beta} = \mathbb{E}\left(Q_b(x)^\alpha y^\beta\right), \qquad \xi_{\alpha,\beta} = \mathbb{E}\left(Q_b(x)^\alpha x^\beta\right). \tag{9}$$

That is, $\xi_{1,1} = \mathbb{E}(Q_b(x)x)$. Note that $\xi_{\alpha,\beta}$ can be represented by $\gamma_{\alpha,\beta}$, but we use both for convenience. Also note that $\xi_{1,1} = \xi_{2,0} = 1 - D_b$ from definitions. For $b = 1, 2, 3, 4, \infty$, we can compute $\xi_{1,1} = 0.6366, 0.8825, 0.9655, 0.9905, 1$, respectively (keeping four decimal points). In fact, it is also known that $D_b = \frac{3^{3/2}2\pi}{12}2^{-2b}$, i.e., the bias decays at the rate of $O(2^{-2b})$. In the following, Theorem 1 summarizes the expectations and variances of the two estimators $\hat{\rho}_{b,f}$ and $\hat{\rho}_{b,f}^{db}$.

**Theorem 1.**

$$\mathbb{E}(\hat{\rho}_{b,f}) = \xi_{1,1}\rho, \qquad \mathbb{E}(\hat{\rho}_{b,f}^{db}) = \rho, \tag{10}$$

$$Var(\hat{\rho}_{b,f}) = \frac{V_{b,f}}{k}, \text{ with } V_{b,f} = (\xi_{2,2} - \xi_{2,0} - \xi_{1,1}^2)\rho^2 + \xi_{2,0} \tag{11}$$

$$Var(\hat{\rho}_{b,f}^{db}) = \frac{V_{b,f}^{db}}{k}, \text{ with } V_{b,f}^{db} = \frac{(\xi_{2,2} - \xi_{2,0} - \xi_{1,1}^2)\rho^2 + \xi_{2,0}}{\xi_{1,1}^2}. \tag{12}$$

**Normalized Estimator.** We also attempt to take advantage of the (beneficial) effect of normalization by defining two normalized estimators and their variances, as summarized in Theorem 2.

**Theorem 2.** *As $k \to \infty$, we have*

$$\hat{\rho}_{b,f,n} = \frac{\sum_{i=1}^{k} Q_b(x_i)y_i}{\sqrt{\sum_{i=1}^{k} Q_b^2(x_i)}\sqrt{\sum_{i=1}^{k} y_i^2}}, \qquad \mathbb{E}(\hat{\rho}_{b,f,n}) = \sqrt{\xi_{1,1}}\rho + O(\frac{1}{k}), \tag{13}$$

$$\hat{\rho}_{b,f,n}^{db} = \frac{\hat{\rho}_{b,f,n}}{\sqrt{\xi_{1,1}}}, \qquad \mathbb{E}(\hat{\rho}_{b,f,n}^{db}) = \rho + O(\frac{1}{k}), \tag{14}$$

$$Var(\hat{\rho}_{b,f,n}) = \frac{V_{b,f,n}}{k} + O(\frac{1}{k^2}), \qquad Var(\hat{\rho}_{b,f,n}^{db}) = \frac{V_{b,f,n}^{db}}{k} + O(\frac{1}{k^2}), \tag{15}$$

$$V_{b,f,n} = \left(\frac{\gamma_{4,0}}{4\gamma_{2,0}} + \frac{3}{4}\gamma_{2,0} + \frac{1}{2}\gamma_{2,2}\right)\rho^2 - \left(\frac{\gamma_{3,1}}{\gamma_{2,0}} + \gamma_{1,3}\right)\rho + \frac{\gamma_{2,2}}{\gamma_{2,0}}, \qquad V_{b,f,n}^{db} = \frac{V_{b,f,n}}{\xi_{1,1}}. \tag{16}$$

### 3.1 Benefits of normalized estimators and debiased estimators

Figure 1 plots (in the left two panels) the variances for two debiased estimators $\hat{\rho}_{b,f}^{db}$ and $\hat{\rho}_{b,f,n}^{db}$, to illustrate the benefits of normalization. The right panel of Figure 1 demonstrates that the variance of the normalized estimator is always smaller, and substantially so as $\rho$ away from zero.

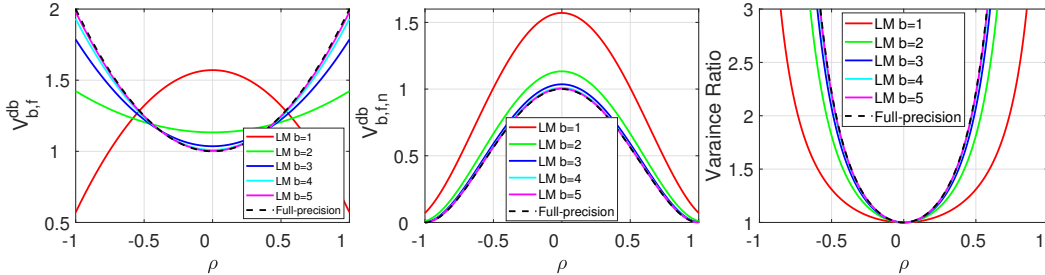

Figure 1: Scenario 1: Comparisons of theoretical variances between two (debiased) estimators $\hat{\rho}_{b,f}^{db}$ and $\hat{\rho}_{b,f,n}^{db}$. **Left panel**: the variance factor $V_{b,f}^{db}$ for $b = 1, 2, 3, 4, 5, \infty$. **Middle panel**: the variance factor $V_{b,f,n}^{db}$ (for the normalized estimator). **Right panel**: the variance ratio: $\frac{V_{b,f}^{db}}{V_{b,f,n}^{db}}$.

To elaborate on the benefit of debiased estimators, we evaluate the mean square errors (MSE): bias$^2$ + variance. Given the benefit of normalization, we consider the two normalized estimators:

$$MSE(\hat{\rho}_{b,f,n}) = \left(1 - \sqrt{\xi_{1,1}}\right)^2 \rho^2 + \frac{V_{b,f,n}}{k} + O(\frac{1}{k^2}), \qquad MSE(\hat{\rho}_{b,f,n}^{db}) = \frac{V_{b,f,n}}{\xi_{1,1}k} + O(\frac{1}{k^2}).$$

Thus, to compare their mean square errors, we can examine the ratio: $\xi_{1,1} + k\rho^2\frac{\xi_{1,1}(1-\sqrt{\xi_{1,1}})^2}{V_{b,f,n}}$, which will be larger than 1 quickly as $k$ increases. Note that $\xi_{1,1} \leq 1$ but it is very close to 1 when $b \geq 3$. In summary, the MSE of the debiased estimator quickly becomes smaller as $k$ increases.

## 3.2 Analysis of mis-ordering probabilities in similarity search

In similarity search, the estimates of inner products are subsequently used for ordering data vectors to identify the nearest neighbor for a given query. Intuitively, a more accurate estimator should provide a more accurate ordering, but a precise analysis is needed for the "mis-ordering" probabilities.

**Definition 2.** *Suppose $u_1, u_2, u_3 \in \mathbb{R}^d$ are three data points (with $u_1$ being a query) with unit norm and pair-wise cosine similarity $\rho_{12}, \rho_{13}$ and $\rho_{23}$ respectively. For an estimator $\hat{\rho}$, the probability of* **mis-ordering** *is defined as*

$$P_{\mathcal{M}}(u_1; u_2, u_3) = Pr\left(\hat{\rho}_{12} > \hat{\rho}_{13} | \rho_{12} < \rho_{13}\right).$$

Consider a case where $u_3$ is the nearest point of $u_1$ in the data space (which implies $\rho_{12} < \rho_{13}$). If the estimation gives $\hat{\rho}_{12} > \hat{\rho}_{13}$, we then make a wrong decision that $u_3$ is not the nearest neighbor of $u_1$.

**Theorem 3.** *(Asymptotic mis-ordering) Suppose $u_1, u_2, u_3 \in \mathbb{R}^d$ are three data points (with $u_1$ being the query) on a unit sphere with pair-wise inner products $\rho_{12}, \rho_{13}$ and $\rho_{23}$ respectively. Denote two estimators $\hat{\rho}$ and $\hat{\rho}'$ based on $k$ random projections such that as $k \to \infty$, the normality $\hat{\rho} \sim N(\alpha\rho, \hat{\sigma}_\rho^2)$ and $\hat{\rho}' \sim N(\alpha'\rho, \hat{\sigma}_\rho'^2)$ hold, with constants $\alpha, \alpha' > 0$. Denote $\delta_\rho^2 = \frac{\hat{\sigma}_\rho^2}{\alpha^2}$, $\delta_\rho'^2 = \frac{\hat{\sigma}_\rho'^2}{\alpha'^2}$ and the correlations $C = corr(\hat{\rho}_{12}, \hat{\rho}_{13})$, $C' = corr(\hat{\rho}'_{12}, \hat{\rho}'_{13})$, respectively. If*

$$\delta'_{\rho_{12}} = a\delta_{\rho_{12}}, \quad \delta'_{\rho_{13}} = a'\delta_{\rho_{13}}, \quad C - aa'C' < \frac{(1-a^2)\delta_{\rho_{12}}^2 + (1-a'^2)\delta_{\rho_{13}}^2}{2\delta_{\rho_{12}}\delta_{\rho_{13}}}, \quad (17)$$

*with some $0 < a < 1$, $0 < a' < 1$, then as $k \to \infty$ we have $\hat{P}_{\mathcal{M}}(u_1; u_2, u_3) > \hat{P}'_{\mathcal{M}}(u_1; u_2, u_3)$, where $\hat{P}_{\mathcal{M}}(u_1; u_2, u_3)$ and $\hat{P}'_{\mathcal{M}}(u_1; u_2, u_3)$ are the mis-ordering probability of $\hat{\rho}$ and $\hat{\rho}'$, respectively.*

**Remark.** *There is an interesting connection with the variances of the aforementioned "debiased estimators". Condition (17) basically assumes that the variance of the debiased $\hat{\rho}'$ is smaller than that of the debiased $\hat{\rho}$ at $\rho_{12}$ and $\rho_{13}$ respectively by factors $a$ and $a'$. In a special case where $a = a'$ and $C = C'$, the last constraint in (17) reduces to $C < \frac{\delta_{\rho_{12}}^2 + \delta_{\rho_{13}}^2}{2\delta_{\rho_{12}}\delta_{\rho_{13}}}$, which always holds since the right-hand side is greater than 1. Also, note that, by Central Limit Theorem, the normality assumption is true for all the estimators we have discussed in this paper.*

Although Theorem 3 is asymptotic, we are able to obtain valuable insights in finite sample case, since statistically a sufficiently large $k$ is enough to provide good approximation to the normal distribution. The important message given by Theorem 3 is that estimators with lower "debiased variance" ($\delta$) tend to have lower mis-ordering probability, which leads to a more accurate estimation of nearest neighbors in the original data space. This could be extremely feasible in numerous real world applications.

## 4 Scenario 2: Quantization with Different Bits

We now consider the more general case (Scenario 2) where the data vectors are LM quantized with different numbers of bits. That is, given observations $\{x_i, y_i\}$, $1 \le i \le n$, we quantize $x_i$ using $b_1$ bits and $y_i$ using $b_2$ bits. Without loss of generality, we assume $b_1 < b_2$. Furthermore, we denote two Lloyd-Max quantizers as $Q_{b_1}$ and $Q_{b_2}$ and distortion $D_{b_1}$ and $D_{b_2}$, respectively. Similar to Scenario 1, we define the asymmetric estimator and the corresponding normalized estimator as

$$\hat{\rho}_{b_1,b_2} = \frac{1}{k}\sum_{i=1}^{k} Q_{b_1}(x_i)Q_{b_2}(y_i), \quad \hat{\rho}_{b_1,b_2,n} = \frac{\sum_{i=1}^{k} Q_{b_1}(x_i)Q_{b_2}(y_i)}{\sqrt{\sum_{i=1}^{k} Q_{b_1}^2(x_i)}\sqrt{\sum_{i=1}^{k} Q_{b_2}^2(y_i)}}. \quad (18)$$

As one might expect, the analysis will become somewhat more difficult. Similar to the analysis for Scenario 1, in this section we will use the following notations:

$$\xi_{\alpha,\beta} = \mathbb{E}\left(Q_{b_1}(x)^\alpha x^\beta\right), \quad \gamma_{\alpha,\beta} = \mathbb{E}\left(Q_{b_2}(x)^\alpha x^\beta\right), \quad \zeta_{\alpha,\beta} = \mathbb{E}\left(Q_{b_1}(x)^\alpha Q_{b_2}(y)^\beta\right), \quad (19)$$

which allow us to express the expectation and variance of $\hat{\rho}_{b_1,b_2}$ as follows.

$$\mathbb{E}\left(\hat{\rho}_{b_1,b_2}\right) = \zeta_{1,1}, \quad Var\left(\hat{\rho}_{b_1,b_2}\right) = \frac{V_{b_1,b_2}}{k}, \quad V_{b_1,b_2} = \zeta_{2,2} - \zeta_{1,1}^2 \quad (20)$$

$\zeta_{1,1}$ can be expressed as an infinite sum, but it appears difficult to be further simplified. Nevertheless, we are able to quantify the expectation of $\hat{\rho}_{b_1,b_2}$ in Theorem 4.

**Theorem 4.** *The following two bounds hold for $\rho \in [-1, 1]$:*

$$\left| \mathbb{E}\left(\hat{\rho}_{b_1,b_2}\right) - (1 - D_{b_1})(1 - D_{b_2})\rho \right| \leq \Delta_1, \quad and \tag{21}$$

$$\Delta_2 - \Delta_1 \leq \left| E\left(\hat{\rho}_{b_1,b_2}\right) - \rho \right| \leq \Delta_1 + \Delta_2, \quad where \tag{22}$$

$$\Delta_1 = \sqrt{D_{b_1} D_{b_2}} \sqrt{1 - D_{b_1}} \sqrt{1 - D_{b_2}} |\rho|^3, \quad \Delta_2 = (D_{b_1} + D_{b_2} - D_{b_1} D_{b_2})|\rho|.$$

**Remark.** *When $b_2 \to \infty$ (i.e., Scenario 1), we have $D_{b_2} \to 0$ and the bound reduces to an equality $\mathbb{E}\left(\hat{\rho}_{b_1,\infty}\right) = (1 - D_{b_1})\rho$, which matches the result in Section 3.*

Eq. (22) provides upper and lower bounds for the absolute bias of $\hat{\rho}_{b_1,b_2}$. When $b_1 = b_2$ (i.e., the symmetric quantization case), Theorem 5 presents more refined bounds of the bias of $\hat{\rho}_{b_1,b_2}$.

**Theorem 5.** *(Symmetric quantization) Suppose $b_1 = b_2 = b$. For $\rho \in [-1, 1]$, we have*

$$(2D_b - D_b^2)|\rho| - D_b(1 - D_b)|\rho|^3 \leq \left| \mathbb{E}\left(\hat{\rho}_{b,b}\right) - \rho \right| \leq (2D_b - D_b^2)|\rho|. \tag{23}$$

**Remark.** *Compared to [29], which derived $|\mathbb{E}(\hat{\rho}_{b,b}) - \rho| \leq 2D_b|\rho|$, our bounds are more tight.*

What about the debiased estimator of $\hat{\rho}_{b_1,b_2}$? It is slightly tricky because $\mathbb{E}(\hat{\rho}_{b_1,b_2}) = \zeta_{1,1}$ cannot be explicitly expressed as $c\rho$ for some constant $c$ (otherwise the debiased estimator would be simply $\hat{\rho}_{b_1,b_2}/c$). In Theorem 4, Eq. (21) implies that the expectation of $\hat{\rho}_{b_1,b_2}$ is close to $(1 - D_{b_1})(1 - D_{b_2})\rho$. Thus, we recommend $\frac{\hat{\rho}_{b_1,b_2}}{(1 - D_{b_1})(1 - D_{b_2})}$ as the surrogate for the debiased estimator.

Next, we provide the expectation and variance of the normalized estimator in Theorem 6.

**Theorem 6.** *(Normalized estimator) As $k \to \infty$, we have*

$$\mathbb{E}\left(\hat{\rho}_{b_1,b_2,n}\right) = \frac{\zeta_{1,1}}{\sqrt{\xi_{2,0}\gamma_{2,0}}} + O\left(\frac{1}{k}\right), \qquad Var\left(\hat{\rho}_{b_1,b_2,n}\right) = \frac{V_{b_1,b_2,n}}{k} + O\left(\frac{1}{k^2}\right), \tag{24}$$

$$V_{b_1,b_2,n} = \frac{\zeta_{2,2} - \zeta_{1,1}^2}{\xi_{2,0}\gamma_{2,0}} - \frac{\zeta_{1,1}\zeta_{3,1} - \zeta_{1,1}^2\xi_{2,0}}{\xi_{2,0}^2\gamma_{2,0}} - \frac{\zeta_{1,1}\zeta_{1,3} - \zeta_{1,1}^2\gamma_{2,0}}{\xi_{2,0}\gamma_{2,0}^2} \tag{25}$$
$$+ \frac{\zeta_{1,1}^2\zeta_{2,2} - \zeta_{1,1}^2\xi_{2,0}\gamma_{2,0}}{2\xi_{2,0}^2\gamma_{2,0}^2} + \frac{\zeta_{1,1}^2\xi_{4,0} - \zeta_{1,1}^2\xi_{2,0}^2}{4\xi_{2,0}^3\gamma_{2,0}} + \frac{\zeta_{1,1}^2\gamma_{4,0} - \zeta_{1,1}^2\gamma_{2,0}^2}{4\xi_{2,0}\gamma_{2,0}^3}.$$

**Remark.** *When $b_2 = \infty$, the expected value of $\hat{\rho}_{b_1,b_2,n}$ reduces to that of $\hat{\rho}_{b_1,f,n}$ in Scenario 1. Additionally, we have $\zeta_{1,1} = \zeta_{2,0}\rho$, $\gamma_{2,0} = 1$, and $\gamma_{4,0} = 3$. It is easy to check that the expression of the variance will reduce to the corresponding formula in Theorem 2. Also, note that $\xi_{2,0} = 1 - D_{b_1}$, $\gamma_{2,0} = 1 - D_{b_2}$, and $\zeta_{1,1} \approx (1 - D_{b_1})(1 - D_{b_2})\rho$. This means that we can practically use $\frac{\hat{\rho}_{b_1,b_2,n}}{\sqrt{(1 - D_{b_1})(1 - D_{b_2})}}$ as surrogate for the debiased estimator of $\hat{\rho}_{b_1,b_2,n}$.*

We plot the related results in Figure 2, which verifies the theories in Theorems 4, 5 and 6.

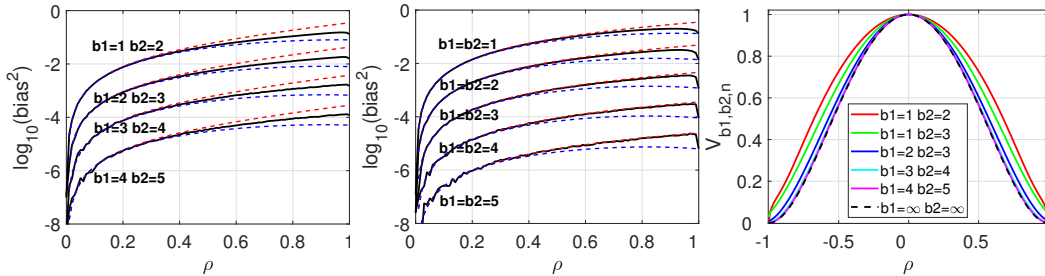

Figure 2: **Left panel**: the absolute bias (solid curves, in $\log_{10}$ scale) of $\hat{\rho}_{b_1,b_2}$ by simulations, along with the upper bound (red dashed curves) and lower bound (blue dashed curves) in Eq. (22). **Middle panel**: the absolute bias of $\hat{\rho}_{b_1,b_2}$ with $b_1 = b_2$ (the symmetric case) along with the upper and lower bounds in Eq. (23). **Right panel**: The variance $V_{b_1,b_2,n}$ of the normalized estimator in Theorem 6.

## 5 Monotonicity of Inner Product Estimates

In applications such as nearest neighbor retrieval, the order of distances tends to matter more than the exact values. Given an estimator $\hat{\rho}$, one would hope that $\mathbb{E}(\hat{\rho})$ is monotone in $\rho$. This is indeed

the case in the full-precision situation. Recall that, in Section 2, given i.i.d. observations $\{x_i, y_i\}$, $i = 1, 2, ...k$, the full-precision estimator $\hat{\rho}_f = \frac{1}{k} \sum_{i=1}^{k} x_i y_i$ is monotone in $\rho$ in the expectation because $\mathbb{E}(\hat{\rho}_f) = \rho$. Naturally, one will ask if the expectations of our quantized estimators, e.g., $\hat{\rho}_{b_1, b_2} = \frac{1}{k} \sum_{i=1}^{k} Q_{b_1}(x_i) Q_{b_2}(y_i)$, are also monotone in $\rho$. This turns out to be non-trivial question.

We solve this important problem rigorously through several stages. Our analysis will not be restricted to LM quantizers. To do so, we will need the following definition about "increasing quantizer".

**Definition 3.** *(Increasing quantizer) Let $Q$ be an $M$-level quantizer with boarders $t_0 < \cdots < t_M$ and reconstruction levels $\mu_1, ..., \mu_M$. We say that $Q$ is an increasing quantizer if $\mu_1 < \cdots < \mu_M$.*

To proceed, we will prove the following three Lemmas for increasing quantizers.

**Lemma 1.** *(1-bit vs. others) Suppose $Q_{b_1}, Q_{b_2}$ are increasing quantizers symmetric about 0, with $b_1 \geq 1$, and $b_2 = 1$. Then $\mathbb{E}(Q_{b_1}(x) Q_{b_2}(y))$ is strictly increasing in $\rho$ on $[-1, 1]$.*

**Lemma 2.** *(2-bit vs. 2-bit) Suppose $Q_{b_1}, Q_{b_2}$ are any two increasing quantizers symmetric about 0, with $b_1 = b_2 = 2$. Then $\mathbb{E}(Q_{b_1}(x) Q_{b_2}(y))$ is strictly increasing in $\rho$ on $[-1, 1]$.*

**Lemma 3.** *(Universal decomposition) For any increasing discrete quantizer $Q_b$, $b \geq 3$ which is symmetric about 0, there exist a 2-bit symmetric increasing quantizer $Q_2$ and a (b-1)-bit symmetric increasing quantizer $Q_{b-1}$ such that $Q_b = Q_{b-1} + Q_2$.*

Once we have the above lemmas, we are ready to prove the monotonicity of $\mathbb{E}(Q_{b_1}(x) Q_{b_2}(y))$.

**Theorem 7.** *(Monotonicity) For any increasing quantizers $Q_{b_1}$ and $Q_{b_2}$ symmetric about 0 with bits $b_1 \geq 1$ and $b_2 \geq 1$, the function $\mathbb{E}(Q_{b_1}(x) Q_{b_2}(y))$ is increasing in $\rho$.*

*Proof.* By Lemma 1, we know that the statement is valid for $b_1 = 1$, and arbitrary $b_2$. Now we look at the case where $b_1 \geq 2, b_2 \geq 2$. By Lemma 3, we can always write

$$Q_{b_1}(x) = \sum_{i=1}^{b_1-1} \tilde{Q}_2^{(i)}(x), \quad Q_{b_2}(y) = \sum_{j=1}^{b_2-1} \hat{Q}_2^{(j)}(y),$$

where $\tilde{Q}_2^1, ..., \tilde{Q}_2^{b_1-1}$ and $\hat{Q}_2^1, ..., \hat{Q}_2^{b_2-1}$ are two sets of symmetric increasing 2-bit quantizers. Thus,

$$\frac{\partial \mathbb{E}(Q_{b_1}(x) Q_{b_2}(y))}{\partial \rho} = \frac{\partial \mathbb{E}(\sum_{i=1}^{b_1-1} \tilde{Q}_2^{(i)}(x) \sum_{j=1}^{b_2-1} \hat{Q}_2^{(j)}(y))}{\partial \rho}$$

$$= \sum_{i=1}^{b_1-1} \sum_{j=1}^{b_2-1} \frac{\partial \mathbb{E}(\tilde{Q}_2^{(i)}(x) \hat{Q}_2^{(j)}(y))}{\partial \rho} > 0,$$

where the last equality is due to linearity of expectation and derivative, and the inequality holds because of Lemma 2. Therefore, $\mathbb{E}(Q_{b_1}(x) Q_{b_2}(y))$ is increasing in $\rho$ for any $b_1 \geq 1$ and $b_2 \geq 1$. $\square$

Recall that, in Section 3.2, we have proved the result for the mis-ordering probability, i.e., Theorem 3, which actually assumes estimators have expectations monotone in $\rho$. Therefore, Theorem 7 provides the necessary proof to support the assumption in Theorem 3.

# 6 Empirical Study: Similarity Search

In this section, we test proposed estimators on 3 datasets from the UCI repository (Table 1) [16]. The experiments clearly confirm that the normalization step uniformly improves the search accuracy. The results also, to an extent, illustrate the influence of mis-ordering probability studied in Theorem 3.

Table 1: Datasets used in the empirical study. Mean $\rho$ is the average pair-wise cosine similarity for sample pairs. Mean 1-NN $\rho$ is the average cosine similarity of each point to its nearest neighbor.

| Dataset | # samples | # features | # classes | Mean $\rho$ | Mean 1-NN $\rho$ |
|---------|-----------|------------|-----------|-------------|------------------|
| Arcene | 200 | 10000 | 2 | 0.63 | 0.86 |
| BASEHOCK | 1993 | 4862 | 2 | 0.33 | 0.59 |
| COIL20 | 1440 | 1024 | 20 | 0.61 | 0.93 |

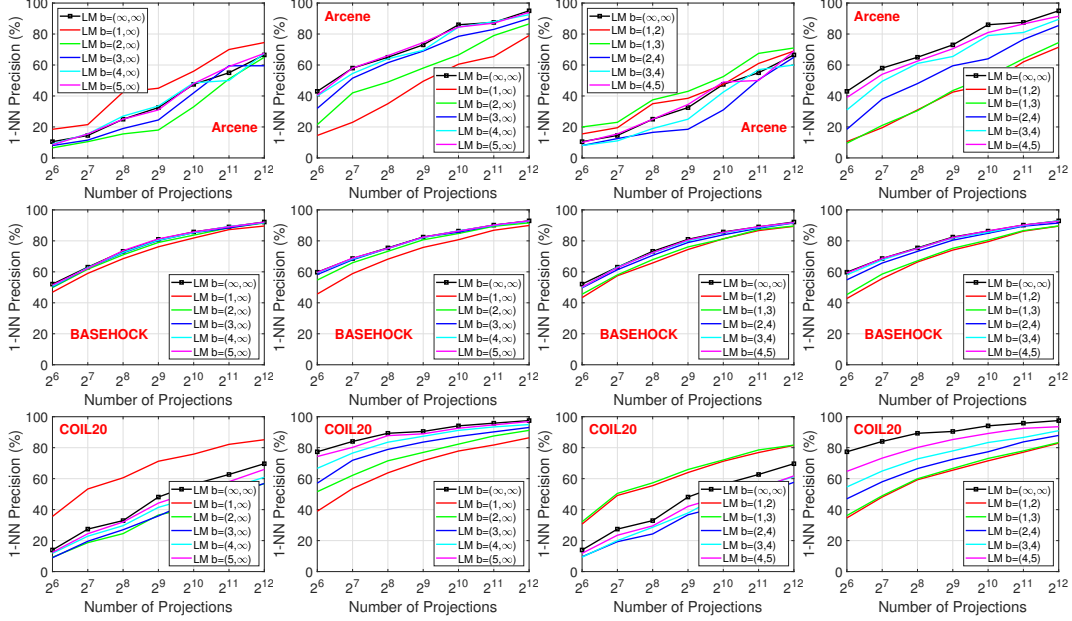

Figure 3: Nearest neighbor search recovery results using cosine similarity and quantized estimators, from random projections. Columns 1 and 2 (Scenario 1): the estimator $\hat{\rho}_{b,f}$ and its normalized version $\hat{\rho}_{b,f,n}$. Columns 3 and 4 (Scenario 2): the estimator $\hat{\rho}_{b_1,b_2}$ and its normalized version $\hat{\rho}_{b_1,b_2,n}$.

For each dataset, all the examples are preprocessed to have unit norm. The evaluation metric we adopt is the **1-NN precision**, which is the proportion of nearest neighbors (NN) we can recover from the nearest neighbor estimated using quantized random projections, averaged over all the examples.

We summarize the results in Figure 3. First of all, we can see that, as the number of bits increases, the performance of the quantized estimators converges to that of the estimator with full-precision, as expected. Importantly, the normalization step of the estimators substantially improves the performances, by comparing Column 2 with Column 1 (for Scenario 1), and Column 4 with Column 3 (for Scenario 2). In addition, we can to an extent validate the assertions in Theorem 3, which states that smaller variance of debiased estimators could improve NN recovery precision.

- In Figure 1 (left panel), we see that the variance of debiased estimate $\hat{\rho}_{b,f}^{db}$ with $b = 1$ is much smaller than using $b \geq 2$ in high similarity region (e.g. $|\rho| > 0.8$), and roughly the same at $\rho = 0.6$. Since *Arcene* and *COIL20* have high mean 1-NN $\rho$ (0.86 and 0.93 respectively), Theorem 3 may imply that cosine estimation of $\hat{\rho}_{1,f}^{db}$ should (in general) have smaller mis-ordering probability than $b \geq 2$, implying higher 1-NN precision. On the other hand, the average 1-NN $\rho$ of *BASEHOCK* is 0.59, so $\hat{\rho}_{b,f}^{db}$ with all $b = 1, 2, ..., \infty$ would likely give similar performance. These claims are consistent with Column 1 of Figure 3.

- The variance of the debiased normalized estimator $\hat{\rho}_{b,f,n}^{db}$ (Figure 1, middle panel) decreases as $b$ increases, uniformly for any $\rho$. Hence by Theorem 3 we expect the 1-NN precision should increase with larger $b$ on all 3 datasets, as confirmed by Column 2 of Figure 3.

# 7 Conclusion

In this paper, we conduct a comprehensive study of estimating inner product similarities from random projections followed by asymmetric quantizations. This setting is theoretically interesting and also has many practical applications. For example, in a retrieval system, data vectors (after random projections) in the repository are quantized to reduce storage and communications; when a new query vector arrives, it does not have to be quantized. Another example of asymmetric quantization may come from data collected from different sources with own quantization strategies. In this study, we propose a series of estimators for asymmetric quantization, starting with the simple basic estimator, then the normalized estimator, and then the debiased estimators. We provide a thorough analysis of the estimation errors. Furthermore, we analyze the problems of "mis-ordering" probabilities and monotonicity properties of estimators. While our methods and analyses are largely based on the classical Lloyd-Max (LM) method, they can be extended to other more general quantization schemes.

## Footnotes

[1] Normalizing each data vector to the unit norm is a standard data preprocessing procedure for many applications such as clustering and classification. In this paper, we adopt this assumption merely for convenience. When data is not normalized, our results still hold, although we will need to store the values of the norms.

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
