[Supplementary Material]

# Supplemental Materials for: Random Projections with Asymmetric Quantization

**Xiaoyun Li**
Department of Statistics
Rutgers University
Piscataway, NJ 08854
xiaoyun.li@rutgers.edu

**Ping Li**
Cognitive Computing Lab
Baidu Research USA
Bellevue, WA 98004
liping11@baidu.com

## A  Proofs of Section 3: Scenario 1

Recall the notations in Section 3,

$$\gamma_{\alpha,\beta} = \mathbb{E}\left(Q_b(x)^\alpha y^\beta\right), \qquad \xi_{\alpha,\beta} = \mathbb{E}\left(Q_b(x)^\alpha x^\beta\right). \tag{1}$$

Also denote $\boldsymbol{x} = (x_1, ..., x_k)$, $\boldsymbol{y} = (y_1, ..., y_k)$, and $Q_b(\boldsymbol{x}) = (Q_b(x_1), ..., Q_b(x_k))$, etc.

The following Lemma is a known result of Lloyd-Max quantizer. We provide a proof here since the proof would be useful for helping readers to better understand the details.

**Lemma A1.** *Let $Q_b$ be a $b$-bit Lloyd-Max quantizer optimized with respect to an arbitrary probability distribution $f$. Suppose random variable $x \sim f$, then*

$$\xi_{1,1} = \xi_{2,0} = 1 - D_b.$$

*Furthermore, if $f$ is standard normal distribution, then $\xi_{1,1} = \xi_{2,0} \leq 1$.*

*Proof.* Recall that each reconstruction level of LM quantizer is the conditional expectations on its corresponding separated region. Let $t_0 < t_1 < \cdots < t_M$ be the boarders. We have

$$
\begin{aligned}
\mathbb{E}(Q_b(x)x) &= \sum_{i=1}^{M} \int_{t_{i-1}}^{t_i} \frac{\int_{t_{i-1}}^{t_i} xf(x)dx}{\int_{t_{i-1}}^{t_i} f(x)dx} xf(x)dx \\
&= \sum_{i=1}^{M} \frac{(\int_{t_{i-1}}^{t_i} xf(x)dx)^2}{\int_{t_{i-1}}^{t_i} f(x)dx} \\
&= \sum_{i=1}^{M} \int_{t_{i-1}}^{t_i} \frac{(\int_{t_{i-1}}^{t_i} xf(x)dx)^2}{(\int_{t_{i-1}}^{t_i} f(x)dx)^2} f(x)dx = \mathbb{E}(Q_b(x)^2).
\end{aligned}
$$

If $f(x) = \phi(x)$ which is standar Gaussian density, we have

$$1 - D_b = 1 - \mathbb{E}((x - Q_b(x)^2)) = 2\mathbb{E}(Q_b(x)x) - \mathbb{E}(Q_b(x)^2) = \xi_{1,1}.$$

The proof is complete. $\qquad\square$

### A.1  Proof of Theorem 1

*Proof.* We have $y_i = \rho x_i + \sqrt{1 - \rho^2}Z$ in distribution, with $Z \sim N(0, 1)$ independent of $x$. Hence,

$$
\begin{aligned}
\mathbb{E}(\hat{\rho}_{b,f}) = \gamma_{1,1} &= \mathbb{E}(Q_b(x)(\rho x_i + \sqrt{1 - \rho^2}Z)) \\
&= \rho\mathbb{E}(Q_b(x_i)x_i) \\
&= \xi_{1,1}\rho.
\end{aligned}
$$

Moreover, we have

$$\begin{aligned}
\gamma_{2,2} &= \mathbb{E}(Q_b(x)^2(\rho x_i + \sqrt{1-\rho^2}Z)^2) \\
&= \xi_{2,2}\rho^2 + (1-\rho^2)\xi_{2,0} \\
&= (\xi_{2,2} - \xi_{2,0})\rho^2 + \xi_{2,0}.
\end{aligned}$$

Therefore, the variance can be expressed as

$$\begin{aligned}
Var(\hat{\rho}_{b,f}) = \frac{1}{k}Var(Q_b(x_i)y_i) &= \frac{1}{k}(\mathbb{E}(Q_b(x_i)^2 y_i^2) - \mathbb{E}(Q_b(x_i)y_i)^2) \\
&= \frac{1}{k}(\gamma_{2,2} - \xi_{1,1}^2\rho^2) \\
&= \frac{(\xi_{2,2} - \xi_{2,0} - \xi_{1,1}^2)\rho^2 + \xi_{2,0}}{k}.
\end{aligned}$$

The variance of debiased estimator follows easily. The proof is complete. $\qquad\square$

## A.2  Proof of Theorem 2

*Proof.* Using first order Taylor expansion of $\frac{x}{y}$ at $x_0, y_0$ we get

$$\frac{x}{y} = \frac{x_0}{y_0} + \frac{x-x_0}{y_0} - \frac{(y-y_0)x_0}{y_0^2} + O(\frac{(y-y_0)^2}{y_0^3}). \tag{2}$$

Therefore,

$$\mathbb{E}(\hat{\rho}_{b,f,n}) = \mathbb{E}\left(\frac{\frac{1}{k}\langle Q_b(\boldsymbol{x}), \boldsymbol{y}\rangle}{\sqrt{\frac{1}{k^2}\|Q_b(\boldsymbol{x})\|_2^2\|\boldsymbol{y}\|_2^2}}\right) = \frac{\mathbb{E}(\hat{\rho}_{b,f})}{\mathbb{E}\left(\sqrt{\frac{1}{k^2}\|Q_b(\boldsymbol{x})\|_2^2\|\boldsymbol{y}\|_2^2}\right)} + O(\frac{1}{k}).$$

Let $T = \frac{1}{k^2}\|Q_b(\boldsymbol{x})\|_2^2\|\boldsymbol{y}\|_2^2$ and $\mathbb{E}(T) = E_0$. Using another Taylor expansion we have:

$$\begin{aligned}
\mathbb{E}(\sqrt{T}) &= \mathbb{E}[\sqrt{E_0} + \frac{T-E_0}{2\sqrt{E_0}} + O((T-E_0)^2)] \\
&= \sqrt{E_0} + O(\frac{1}{k}), \ as \ k \to \infty,
\end{aligned}$$

and

$$\begin{aligned}
E_0 = \mathbb{E}(T) &= \frac{1}{k^2}\mathbb{E}[(\sum_i^k Q_b(x_i)^2)(\sum_i^k y_i^2)] \\
&= \frac{1}{k^2}(\mathbb{E}[\sum_{i\neq j} Q_b(x_i)^2 y_j^2] + \mathbb{E}[\sum_{l=1}^k Q(x_l)^2 y_l^2]) \\
&= \frac{k(k-1)}{k^2}\mathbb{E}(Q(x_1)^2) + \frac{1}{k}\mathbb{E}(Q(x_1)^2 y_1^2) \\
&= \frac{k-1}{k}\xi_{2,0} + \frac{\gamma_{2,2}}{k} + O(\frac{1}{k}), \ as \ k \to \infty.
\end{aligned}$$

Put above parts together, we obtain the expected value as $k \to \infty$,

$$\mathbb{E}(\hat{\rho}_{b,f,n}) = \frac{\xi_{1,1}\rho}{\sqrt{\xi_{2,0}}} + O(\frac{1}{k}).$$

To derive the asymptotic variance, let $a = \frac{\langle Q_b(x),y\rangle}{k}$, $b = \frac{\|Q_b(x)\|^2}{k}$, $c = \frac{\|y\|^2}{k}$, and hence $\hat{\rho}_{b,f,n} = \frac{a}{\sqrt{b}\sqrt{c}}$.

We have

$$\mathbb{E}(a) = \xi_{1,1}\rho = \xi_{2,0}\rho = \gamma_{2,0}\rho, \ Var(a) = \frac{\gamma_{2,2} - \gamma_{2,0}^2\rho^2}{k},$$

$$\mathbb{E}(b) = \xi_{2,0} = \gamma_{2,0}, Var(b) = \frac{\gamma_{4,0} - \gamma_{2,0}^2}{k},$$

$$\mathbb{E}(c) = 1, Var(c) = \frac{2}{k},$$

$$Cov(a,b) = \mathbb{E}\left[\frac{1}{k^2}(\sum_1^k Q_b(x_i)y_i)(\sum_1^k Q_b(x_i)^2)\right] - \mathbb{E}(a)\mathbb{E}(b),$$

$$= \frac{1}{k^2}[k(k-1)\gamma_{2,0}\cdot\gamma_{2,0}\rho + k\gamma_{3,1}] - \gamma_{2,0}^2\rho,$$

$$= \frac{\gamma_{3,1} - \gamma_{2,0}^2\rho}{k}.$$

Similarly, we can get

$$Cov(a,c) = \frac{\gamma_{1,3} - \gamma_{2,0}\rho}{k}, \quad Cov(b,c) = \frac{\gamma_{2,2} - \gamma_{2,0}}{k}.$$

Hence the covariance matrix is formulated as

$$Cov(a,b,c) = \frac{1}{k}\begin{pmatrix} \gamma_{2,2} - \gamma_{2,0}^2\rho^2 & \gamma_{3,1} - \gamma_{2,0}^2\rho & \gamma_{1,3} - \gamma_{2,0}\rho \\ \gamma_{3,1} - \gamma_{2,0}^2\rho & \gamma_{4,0} - \gamma_{2,0}^2 & \gamma_{2,2} - \gamma_{2,0} \\ \gamma_{1,3} - \gamma_{2,0}\rho & \gamma_{2,2} - \gamma_{2,0} & 2 \end{pmatrix},$$

and the gradients

$$\triangledown(a,b,c) = (\frac{1}{\sqrt{bc}}, -\frac{a}{2b^{\frac{3}{2}}\sqrt{c}}, -\frac{a}{2c^{\frac{3}{2}}\sqrt{b}}).$$

Second order Taylor expansion gives

$$Var(\hat{\rho}_{b,f,n}) = \triangledown(\mathbb{E}(a),\mathbb{E}(b),\mathbb{E}(c))^T Cov(a,b,c)\triangledown(\mathbb{E}(a),\mathbb{E}(b),\mathbb{E}(c)) + O(\frac{1}{k^2}),$$

and the final result is derived by plugging in the expressions and collecting terms:

$$Var(\hat{\rho}_{b,f,n}) = \frac{1}{k}[(\frac{\gamma_{4,0}}{4\gamma_{2,0}} + \frac{3}{4}\gamma_{2,0} + \frac{1}{2}\gamma_{2,2})\rho^2 - (\frac{\gamma_{3,1}}{\gamma_{2,0}} + \gamma_{1,3})\rho + \frac{\gamma_{2,2}}{\gamma_{2,0}}] + O(\frac{1}{k^2}).$$

This concludes the proof. $\qquad\qquad\qquad\qquad\qquad\qquad\qquad\qquad\qquad\qquad\qquad\square$

### A.3 Proof of Theorem 3

*Proof.* By normality assumption, we can compute,

$$\hat{P}_{\mathcal{M}}(u_1, u_2, u_3) = 1 - \Phi(\frac{\alpha(\rho_{12} - \rho_{13})}{\sqrt{\sigma_{\rho_{12}}^2 + \sigma_{\rho_{13}}^2 - 2C\sigma_{\rho_{12}}\sigma_{\rho_{13}}}}),$$

$$\hat{P}'_{\mathcal{M}}(u_1, u_2, u_3) = 1 - \Phi(\frac{\alpha'(\rho_{12} - \rho_{13})}{\sqrt{\sigma'^2_{\rho_{12}} + \sigma'^2_{\rho_{13}} - 2C'\sigma'_{\rho_{12}}\sigma'_{\rho_{13}}}}).$$

We can rewrite in terms of debiased variances by $\sigma_\rho^2 = \delta_\rho^2\alpha^2$ and $\sigma'^2_\rho = \delta'^2_\rho\alpha'^2$ for $\forall\rho$:

$$\hat{P}_{\mathcal{M}}(u_1, u_2, u_3) = 1 - \Phi(\frac{\rho_{12} - \rho_{13}}{\sqrt{\delta_{\rho_{12}}^2 + \delta_{\rho_{13}}^2 - 2C\delta_{\rho_{12}}\delta_{\rho_{13}}}}),$$

$$\hat{P}'_{\mathcal{M}}(u_1, u_2, u_3) = 1 - \Phi(\frac{\rho_{12} - \rho_{13}}{\sqrt{\delta'^2_{\rho_{12}} + \delta'^2_{\rho_{13}} - 2C'\delta'_{\rho_{12}}\delta'_{\rho_{13}}}})$$

$$= 1 - \Phi(\frac{\rho_{12} - \rho_{13}}{\sqrt{a^2\delta_{\rho_{12}}^2 + a'^2\delta_{\rho_{13}}^2 - 2aa'C'\delta_{\rho_{12}}\delta_{\rho_{13}}}}),$$

with $0 < a < 1$, $0 < a' < 1$ by assumption. To compare the probabilities it suffices to consider the denominators. To make $\hat{P}'_{\mathcal{M}}(u_1, u_2, u_3) < \hat{P}_{\mathcal{M}}(u_1, u_2, u_3)$, we need

$$\delta^2_{\rho_{12}} + \delta^2_{\rho_{13}} - 2C\delta_{\rho_{12}}\delta_{\rho_{13}} > a^2\delta^2_{\rho_{12}} + a'^2\delta^2_{\rho_{13}} - 2aa'C'\delta_{\rho_{12}}\delta_{\rho_{13}},$$

which after some simplification gives the condition

$$C - aa'C' < \frac{(1-a^2)\delta^2_{\rho_{12}} + (1-a'^2)\delta^2_{\rho_{13}}}{2\delta_{\rho_{12}}\delta_{\rho_{13}}}.$$

The proof is complete. $\qquad\square$

## B Proofs of Section 4: Scenario 2 & Symmetric quantization

**Hermite polynomials.** First we introduce an important tool for our following analysis. The probabilists' Hermite polynomials are defined as

$$H_l(x) = (-1)^l \exp(\frac{x^2}{2}) \frac{d^l}{dx^l} \exp(-\frac{x^2}{2}),$$

which form an orthogonal basis of the Hilbert space $\mathscr{H}$ of all functions satisfying $\int |f(x)|^2 e^{-\frac{x^2}{2}} dx < \infty$, $w.r.t$ the $e^{-\frac{x^2}{2}}$ measure. The inner product is well-defined as

$$\langle f, g \rangle = \int f(x)g(x)e^{-\frac{x^2}{2}} dx.$$

As an example, the first several Hermite polynomials are

$$H_0(x) = 1, \ H_1(x) = x, \ H_2(x) = x^2 - 1, \ H_3(x) = x^3 - 3x, ...,$$

and they can be derived via a recursion relationship: for $l = 0, 1, ...,$

$$H_{l+1}(x) = xH_l(x) - H'_l(x).$$

Hermite Polynomials admits **Orthogonality** in the sense that

$$\int H_m(x)H_n(x)e^{-\frac{x^2}{2}} dx = 0, \quad m \neq n,$$

$$\int H_n(x)H_n(x)e^{-\frac{x^2}{2}} dx = \sqrt{2\pi}n!, \quad m = n.$$

We can deduct some useful quantities from this property. Let $x \sim N(0,1)$, then we have for all $l = 1, 2, ...,$

$$\mathbb{E}(H_l(x)) = \mathbb{E}(H_0(x)H_l(x)) = 0, \ Var(H_l(x)) = \frac{1}{\sqrt{2\pi}} \int H_l(x)H_l(x)e^{-\frac{x^2}{2}} dx = l!.$$

Moreover, $H_n(x)$ is an odd function if $n$ is odd, and is symmetric about $y$ axis when $n$ is even. One important application of Hermite polynomials is that we can decompose the bivariate normal density as below [1]:

$$\phi_\rho(x, y) = \sum_{l=0}^{\infty} \frac{\rho^l}{l!} H_l(x)H_l(y)\phi(x)\phi(y),$$

where $H_l(x)$ is the $l$-th order probabilitist Hermite polynomial, and $\phi(x)$ is the density function of standard normal distribution as defined before. This immediately implies that for any functions $f_1$ and $f_2$, we can write

$$\begin{aligned}
\mathbb{E}[f_1(x)f_2(y)] &= \int \int f_1(x)f_2(y)\phi_\rho(x, y)dxdy \\
&= \int \int f_1(x)f_2(y)\sum_{l=0}^{\infty} \frac{\rho^l}{l!} H_l(x)H_l(y)\phi(x)\phi(y)dxdy \\
&= \sum_{l=0}^{\infty} \frac{\rho^l}{l!} \int \int f_1(x)f_2(y)H_l(x)H_l(y)\phi(x)\phi(y)dxdy \\
&= \sum_{l=0}^{\infty} \frac{\rho^l}{l!} (\int f_1(x)H_l(x)\phi(x)dx \int f_2(y)H_l(y)\phi(y)dy).
\end{aligned} \quad (3)$$

As we can see, the correlation coefficient $\rho$ is factored out in (3), which is beneficial for studying the dependence of the expected value on $\rho$.

Now we recall some notations. The data vectors are LM quantized with different bits $b_1 < b_2$, and we denote two Lloyd-Max quantizers as $Q_{b_1}$ and $Q_{b_2}$ and distortion $D_{b_1}$ and $D_{b_2}$, respectively. With a little abuse of notation, in this section we re-define $\xi_{\alpha,\beta} = \mathbb{E}(Q_{b_1}(x)^\alpha x^\beta)$, $\gamma_{\alpha,\beta} = \mathbb{E}(Q_{b_2}(x)^\alpha x^\beta)$ and $\zeta_{\alpha,\beta} = \mathbb{E}(Q_{b_1}(x)^\alpha Q_{b_2}(y)^\beta)$.

## B.1  Proof of Theorem 4 & Corollary 1

To prove the results, we will use the following lemma.

**Lemma B2.** *Suppose we have a sequence of positive constants $V = (v_1, v_2, ...)$. Let $W = diag(V)$ and $c_1 = (c_{11}, c_{12}, ...)$ and $c_2 = (c_{21}, c_{22}, ...)$ be vectors with same length as $V$. Then*

$$\max_{\|c_1\|_2^2 = L_1, \|c_2\|_2^2 = L_2} c_1^T W c_2 = \sqrt{L_1 L_2}\|V\|_\infty,$$

*where the infinite norm $\|\cdot\|_\infty$ is the maximum absolute value of a vector.*

*Proof.* By the symmetry of this optimization problem, we know that the optimal solution of $c_1$ and $c_2$ is not unique. Hence, we may cast two more constraints $c_1 \geq 0$ and $c_2 \geq 0$ to get a unique solution. To proceed, we introduce Lagrangian multipliers $L$ with slack variables $\tilde{s} = (s_1, s_2, ...), \tilde{t} = (t_1, t_2, ...)$ as:

$$L = c_1^T W c_2 - \lambda_1(c_1^T c_1 - L_1) - \lambda_2(c_2^T c_2 - L_2) + \tilde{\lambda}_3^T(c_1 - \tilde{s}^2) - \tilde{\lambda}_4^T(c_2 + \tilde{t}^2),$$

where $\tilde{\lambda}_3 = (\lambda_{31}, \lambda_{32}, ...)$ and $\tilde{\lambda}_4 = (\lambda_{41}, \lambda_{42}, ...)$. The Karush-Kuhn-Tucker conditions are satisfied at minimal point, which gives

$$\begin{cases} W c_2 - 2\lambda_1 c_1 + \tilde{\lambda}_3 = 0 & (4) \\ W c_1 - 2\lambda_2 c_2 - \tilde{\lambda}_4 = 0 & (5) \\ c_1^T c_1 = L_1 \\ c_2^T c_2 = L_2 \\ c_1 - \tilde{s}^2 = 0 \\ c_2 + \tilde{t}^2 = 0 \\ 2\tilde{\lambda}_3 \odot \tilde{s} = 0 \\ 2\tilde{\lambda}_4 \odot \tilde{t} = 0 \end{cases}$$

where $\odot$ denotes element-wise product. The equations leads to following observations:

- Any pair of values $(c_{1i}, c_{2i})$ must be zero or nonzero at the same time. To see this, suppose $c_{1i} = 0$ and $c_{2i} \neq 0$, then by (5) we have two situations:
  1) $\lambda_2 \neq 0$ and $\lambda_{4i} \neq 0$, which implies that $t_i = 0$ and thus $c_{2i} = 0$. A contradiction occurs.
  2) $\lambda_2 = 0$ and $\lambda_{4i} = 0$. Firstly, we note that there must exist at least one $j \neq i$ such that $c_{1j} \neq 0$. For a nonzero $c_{1j}$, $\lambda_2 = 0$ forces $\lambda_{4j} \neq 0$, and thus $c_{2j}$ must be zero. Therefore, for $\forall i = 1, 2, ...$, we have $\mathbb{1}\{c_{1i} > 0\} + \mathbb{1}\{c_{2i} > 0\} \leq 1$, which implies that the objective function is trivially 0. Hence it can not be an optimal solution.

- If $c_{1i} \neq 0, c_{2i} \neq 0$ for a $i \in$, then $\lambda_{3i} = \lambda_{4i} = 0$ for $\forall i$. From (4) and (5) we deduct that $c_{1i} = \frac{\lambda_2 c_{2i}}{V_i} = \frac{V_i c_{2i}}{\lambda_1}$, from which we can further derive $V_i^2 = \lambda_1 \lambda_2$.

Based on above reasoning, we can consider 2 situations for the optimal solution. First, if only one pair $(c_{1i}, c_{2i})$ is nonzero, then the maximum of $c_1^T W c_2$ is trivially derived at

$$c_1 = \sqrt{L_1} I_{max}, \ c_2 = \sqrt{L_2} I_{max},$$

with $\boldsymbol{I_{max}}$ the indicator vector of where the maximum of $\boldsymbol{V}$ is located , $e.g$ in the form $(...,0,0,1,0,...)$. The maxima in this case equals to

$$\max_{\boldsymbol{c_1},\boldsymbol{c_2}} \boldsymbol{c_1}^T \boldsymbol{W} \boldsymbol{c_2} = \sqrt{L_1 L_2} \max \boldsymbol{V} = \sqrt{L_1 L_2} \|\boldsymbol{V}\|_\infty,$$

subject to constraints $\|c_1\|_2^2 = L_1, \|c_2\|_2^2 = L_2$.

Now consider the case where more than two pairs of values $(c_{1i}, c_{2i})$, $i \in \mathcal{S}$ are nonzero, where $\mathcal{S}$ denotes the set of nonzero indices. Then $\lambda_1 \lambda_2 = V_i^2 := V^{*2}, \forall i \in \mathcal{S}$ must hold. By Cauchy-Schwartz inequality, we have

$$\boldsymbol{c_1}^T \boldsymbol{W} \boldsymbol{c_2} = V^* \boldsymbol{c_1}^T \boldsymbol{c_2} \le V^* \|\boldsymbol{c_1}\|_2 \|\boldsymbol{c_2}\|_2 \le \sqrt{L_1 L_2} V^* \le \sqrt{L_1 L_2} \|\boldsymbol{V}\|_\infty,$$

and the bound is tight ($i.e.$ equality holds when $\boldsymbol{c_1}$ and $\boldsymbol{c_2}$ have same direction).

Combining above analysis, we have shown that

$$\max_{\|c_1\|_2^2 = L_1, \|c_2\|_2^2 = L_2} \boldsymbol{c_1}^T W \boldsymbol{c_2} = \sqrt{L_1 L_2} \|V\|_\infty.$$

$\square$

**Proof of Theorem 4 and Theorem 5.**

*Proof.* First, we have that

$$\mathbb{E}(Q_{b_1}(x)Q_{b_2}(y)) \qquad (6)$$

$$= \sum_{l=0}^\infty \frac{\rho^l}{l!} \left( \int Q_{b_1}(x)H_l(x)\phi(x)dx \int Q_{b_2}(y)H_l(y)\phi(y)dy \right)$$

$$= \sum_{l=1,odd}^\infty \frac{\rho^l}{l!} E[Q_{b_1}(x)H_l(x)]E[Q_{b_2}(x)H_l(x)]$$

$$= (1 - D_{b_1} - D_{b_2} + D_{b_1}D_{b_2})\rho + \sum_{l=3,odd}^\infty \frac{\rho^l}{l!} Cov[Q_{b_1}(x), H_l(x)] \cdot Cov[Q_{b_2}(x), H_l(x)]. \qquad (7)$$

Note that $E_{-\rho}[Q_{b_1}(x)Q_{b_2}(y)] = -E_\rho[Q_{b_1}(x)Q_{b_2}(y)]$, so it suffices to consider the case where $\rho \ge 0$ in the remaining part of the proof.

From previous sections we know that for a fixed quantizer $Q_b(\cdot)$ with distortion $D_b$ and Hermite Polynomial $H_k(\cdot)$ with $k > 1$,

$$Var(H_k(x)) = \mathbb{E}(H_k(x)^2) = k!, \;\; Cov(Q_b(x), x) = \mathbb{E}(Q_b(x)x) = 1 - D_b,$$

$$Var(Q_b(x)) = \mathbb{E}(Q_b(x)^2) = 1 - D_b, \;\; Cov(H_k(x), x) = \mathbb{E}(H_k(x)x) = 0.$$

We can compute the correlations:

$$Corr(Q_b(x), x) = \sqrt{1 - D_b}, \;\; Corr(H_k(x), x) = 0.$$

By working with correlations between 3 random variables and using Cauchy-Schwartz inequality, we get

$$-\sqrt{D_b} \le Corr(Q_b(x), H_k(x)) \le \sqrt{D_b}.$$

Denote the correlation $Corr(Q_b(x), H_k(x))$ as $c_k$, $k = 0, 1, 2, ...$, and $C=(c_0, c_1, c_2, ...)$. Note that Hermite polynomials are infinite orthogonal basis of the function space $\mathscr{H}$, and thus we have the decomposition $Q_b(x) = \sum_{i=1}^\infty a_i H_i(x)$ for some $a_i$, $i = 1, 2, ....$ Simple calculation yields $Cov(Q, H_i) = a_i Var(H_i(x)), Var(Q) = \sum_{i=1}^\infty a_i^2 Var(H_i(x))$. So the correlations can be derived as

$$c_i = Corr(Q, H_i) = \frac{a_i Var(H_i(x))}{\sqrt{\sum_{j=1}^\infty a_j^2 Var(H_j(x))}\sqrt{Var(H_i(x))}} = \frac{\sqrt{a_i Var(H_i(x))}}{\sqrt{\sum_{j=1}^\infty a_j^2 Var(H_j(x))}}.$$

Consequently, we have $C^T C \equiv 1$. Given that $c_1 = Corr(Q_b(x), x) = \sqrt{1 - D_b}$ and $c_k = 0$ for all even $k$'s, we have $\sum_{k=3,odd}^{\infty} c_k^2 = D_b$.

The above argument holds for both $Q_{b_1}$ and $Q_{b_2}$. Denote $c_{1k} = Corr(Q_{b_1}, H_k)$ and $c_{2k} = Corr(Q_{b_2}, H_k)$ and notice that for $i = 1, 2$ and $k = 0, 1, 2, ...$,

$$Cov(Q_{b_i}(x), H_k(x)) = c_{ik} \sqrt{1 - D_i} \sqrt{k!},$$

because $Var[H_k(x)] = k!$. Continuing with (7) we obtain

$$\mathbb{E}(\hat{\rho}_{b_1,b_2}) = E(Q_{b_1}(x) Q_{b_2}(y)) = (1 - D_{b_1})(1 - D_{b_2})\rho + \sqrt{1 - D_{b_1}} \sqrt{1 - D_{b_2}} \sum_{k=3,odd}^{\infty} c_{1k} c_{2k} \rho^k. \tag{8}$$

Now we seek to bound the last term in above equation. Applying Lemma B2 with $V(\rho) = (\rho^3, \rho^5, \rho^7, ...)$ and constraints $\|c_1\|_2^2 = D_{b_1}, \|c_2\|_2^2 = D_{b_2}$, we get

$$-\sqrt{D_{b_1} D_{b_2}} |\rho|^3 \leq \sum_{k=3,odd}^{\infty} c_{1k} c_{2k} \rho^k \leq \sqrt{D_{b_1} D_{b_2}} |\rho|^3,$$

for $\rho \in [-1, 1]$ by symmetry, and this bound is tight in worst-case. Therefore, we have

$$|\mathbb{E}(\hat{\rho}_{b_1,b_2}) - (1 - D_{b_1})(1 - D_{b_2})\rho| \leq \sqrt{D_{b_1} D_{b_2}} \sqrt{1 - D_{b_1}} \sqrt{1 - D_{b_2}} |\rho|^3. \tag{9}$$

To get the bound on absolute bias, note that for $\rho > 0$, by Eq.(9) we have

$$\rho - \mathbb{E}(\hat{\rho}_{b_1,b_2}) \geq (D_{b_1} + D_{b_2} - D_{b_1} D_{b_2})\rho - \sqrt{D_{b_1} D_{b_2}} \sqrt{1 - D_{b_1}} \sqrt{1 - D_{b_2}} \rho^3. \tag{10}$$

By computing

$$(D_{b_1} + D_{b_2} - D_{b_1} D_{b_2})^2 - D_{b_1} D_{b_2}(1 - D_{b_1})(1 - D_{b_2})$$
$$= D_{b_1}^2 + D_{b_2}^2 + D_{b_1} D_{b_2}(1 - D_{b_1} - D_{b_2})$$
$$\geq 0,$$

since for LM quantizers, $1 - D_{b_1} - D_{b_2} > 0$ always holds. Consequently, we know that $\rho - \mathbb{E}(\hat{\rho}_{b_1,b_2}) > 0$ for $\rho > 0$. Now by the symmetry of $\hat{\rho}_{b_1,b_2}$ and elementary inequalities, we have for $\rho \in [-1, 1]$,

$$\Delta_2 - \Delta_1 \leq |E(\hat{\rho}_{b_1,b_2}) - \rho| \leq \Delta_1 + \Delta_2,$$

where

$$\Delta_1 = \sqrt{D_{b_1} D_{b_2}} \sqrt{1 - D_{b_1}} \sqrt{1 - D_{b_2}} |\rho|^3, \quad \Delta_2 = (D_{b_1} + D_{b_2} - D_{b_1} D_{b_2})|\rho|.$$

To prove Theorem 5, notice that when $Q_1 = Q_2 := Q_b$ and $D_{b_1} = D_{b_2} := D_b$, we can modify (8) as

$$\mathbb{E}(Q_b(x) Q_b(y)) = (1 - 2D_b + D_b^2)\rho + (1 - D_b) \sum_{l=3,odd}^{\infty} c_k^2 \rho^k,$$

where $c_k = Corr(Q_b(x), H_k(x))$ and $\sum_{k=3}^{\infty} c_k^2 = D_b$. Obviously, the summation is lower bounded by 0 and upper bounded by $D_b \rho^3$. Similar calculation can be conducted to get the bound on absolute bias. This completes the proof.

$\square$

## B.2 Proof of Theorem 6

*Proof.* The proof follows from the proof of Theorem 2. As $k \to \infty$, Taylor Expansion of $\frac{x}{y}$ at $x_0, y_0$ gives:

$$\frac{x}{y} = \frac{x_0}{y_0} + \frac{x - x_0}{y_0} - \frac{(y - y_0)x_0}{y_0^2} + O((x - x_0)^2) + O((y - y_0)^2).$$

We apply the expansion at expectations:

$$\mathbb{E}(\hat{\rho}_{b_1,b_2,n}) = \frac{\mathbb{E}(\frac{1}{k}\langle Q_{b_1}(\boldsymbol{x}), Q_{b_2}(\boldsymbol{y})\rangle)}{\mathbb{E}\left(\sqrt{\frac{1}{k^2}\|Q_{b_1}(\boldsymbol{x})\|_2^2 \|Q_{b_2}(\boldsymbol{y})\|_2^2}\right)} + O(\frac{1}{k}) = \frac{\mathbb{E}(\hat{\rho}_{b_1,b_2})}{\mathbb{E}(\sqrt{T})} + O(\frac{1}{k}).$$

Let $T = \frac{1}{k^2}\|Q_{b_1}(\boldsymbol{x})\|_2^2\|Q_{b_2}(\boldsymbol{y})\|_2^2$, $\mathbb{E}(T) = E_0$. Using Taylor Expansion again, we have:

$$\mathbb{E}(\sqrt{T}) = \mathbb{E}\left(\sqrt{E_0} + \frac{T - E_0}{2\sqrt{E_0}} + O((T - E_0)^2)\right)$$

$$= \sqrt{E_0} + O(\frac{1}{k}), \text{ as } k \to \infty,$$

$$E_0 = \mathbb{E}(T) = \frac{1}{k^2}\mathbb{E}\left((\sum_i^k Q_{b_1}(x_i)^2)(\sum_i^k Q_{b_2}(y_i)^2)\right)$$

$$= \frac{1}{k^2}\left[\mathbb{E}(\sum_{i \neq j} Q_{b_1}(x_i)^2 Q_2(y_j)^2) + \mathbb{E}(\sum_{l=1}^k Q_1(x_l)^2 Q_2(y_l)^2)\right]$$

$$= \frac{k(k-1)}{k^2}\mathbb{E}[Q_1(x_1)^2]\mathbb{E}[Q_2(y_1)^2] + \frac{1}{k}\mathbb{E}[Q_1(x_1)^2 Q_2(y_1)^2]$$

$$= \frac{k-1}{k}\xi_{2,0}\gamma_{2,0} + \frac{1}{k}\zeta_{2,2} + O(\frac{1}{k}), \text{ as } k \to \infty.$$

Combining parts together we have as $k \to \infty$,

$$\mathbb{E}(\hat{\rho}_{b_1,b_2,n}) = \frac{\zeta_{1,1}}{\sqrt{\xi_{2,0}\gamma_{2,0}}} + O(\frac{1}{k}).$$

Let $a = \frac{<Q_{b_1}(x), Q_{b_2}(y)>}{k}$, $b = \frac{\|Q_{b_1}(x)\|^2}{k}$, $c = \frac{\|Q_{b_2}(y)\|^2}{k}$, and thus $\hat{\rho}_{b_1,b_2,n} = \frac{a}{\sqrt{b}\sqrt{c}}$. We have:

$$\mathbb{E}(a) = \zeta_{1,1}, Var(a) = \frac{\zeta_{2,2} - \zeta_{1,1}^2}{k},$$

$$\mathbb{E}(b) = \xi_{2,0}, Var(b) = \frac{\xi_{4,0} - \xi_{2,0}^2}{k},$$

$$\mathbb{E}(c) = \gamma_{2,0}, Var(c) = \frac{\gamma_{4,0} - \gamma_{2,0}^2}{k},$$

$$Cov(a, b) = \mathbb{E}\left(\frac{1}{k^2}(\sum_1^k Q_{b_1}(x_i)Q_{b_2}(y_i))(\sum_1^k Q_{b_1}(x_i)^2)\right) - \mathbb{E}(a)\mathbb{E}(b)$$

$$= \frac{1}{k^2}[k(k-1)\zeta_{1,1}\xi_{2,0} + k\zeta_{3,1}] - \zeta_{1,1}\xi_{2,0}$$

$$= \frac{\zeta_{3,1} - \zeta_{1,1}\xi_{2,0}}{k},$$

$$Cov(a, c) = \frac{\zeta_{1,3} - \zeta_{1,1}\gamma_{2,0}}{k}, \; Cov(b, c) = \frac{\zeta_{2,2} - \xi_{2,0}\gamma_{2,0}}{k}.$$

Therefore,

$$Cov(a, b, c) = \frac{1}{k}\begin{pmatrix} \zeta_{2,2} - \zeta_{1,1}^2 & \zeta_{3,1} - \zeta_{1,1}\xi_{2,0} & \zeta_{1,3} - \zeta_{1,1}\gamma_{2,0} \\ \zeta_{3,1} - \zeta_{1,1}\xi_{2,0} & \xi_{4,0} - \xi_{2,0}^2 & \zeta_{2,2} - \xi_{2,0}\gamma_{2,0} \\ \zeta_{1,3} - \zeta_{1,1}\gamma_{2,0} & \zeta_{2,2} - \xi_{2,0}\gamma_{2,0} & \gamma_{4,0} - \gamma_{2,0}^2 \end{pmatrix},$$

and

$$\nabla(\mathbb{E}(a), \mathbb{E}(b), \mathbb{E}(c)) = (\frac{1}{\sqrt{\xi_{2,0}\gamma_{2,0}}}, -\frac{\zeta_{1,1}}{2\xi_{2,0}^{\frac{3}{2}}\sqrt{\gamma_{2,0}}}, -\frac{\zeta_{1,1}}{2\gamma_{2,0}^{\frac{3}{2}}\sqrt{\xi_{2,0}}}).$$

Using Taylor expansion we have

$$Var(\hat{\rho}_{b_1,b_2,n}) = \nabla(\mathbb{E}(a), \mathbb{E}(b), \mathbb{E}(c))^T Cov(a, b, c)\nabla(\mathbb{E}(a), \mathbb{E}(b), \mathbb{E}(c)) + O(\frac{1}{k^2}).$$

The final result is derived by direct calculation and collecting terms:

$$Var(\hat{\rho}_{b_1,b_2,n}) = \frac{1}{k}[\frac{\zeta_{2,2} - \zeta_{1,1}^2}{\xi_{2,0}\gamma_{2,0}} - \frac{\zeta_{1,1}\zeta_{3,1} - \zeta_{1,1}^2\xi_{2,0}}{\xi_{2,0}^2\gamma_{2,0}} - \frac{\zeta_{1,1}\zeta_{1,3} - \zeta_{1,1}^2\gamma_{2,0}}{\xi_{2,0}\gamma_{2,0}^2}$$

$$+ \frac{\zeta_{1,1}^2\zeta_{2,2} - \zeta_{1,1}^2\xi_{2,0}\gamma_{2,0}}{2\xi_{2,0}^2\gamma_{2,0}^2} + \frac{\zeta_{1,1}^2\xi_{4,0} - \zeta_{1,1}^2\xi_{2,0}^2}{4\xi_{2,0}^3\gamma_{2,0}} + \frac{\zeta_{1,1}^2\gamma_{4,0} - \zeta_{1,1}^2\gamma_{2,0}^2}{4\xi_{2,0}\gamma_{2,0}^3}] + O(\frac{1}{k^2}).$$

This completes the proof. $\qquad\square$

# C  Proofs of Section 5: Monotonicity

## C.1  Proof of Theorem 7

**Lemma C3.** *Assume* $\begin{pmatrix} x \\ y \end{pmatrix} \sim N\left( \begin{pmatrix} 0 \\ 0 \end{pmatrix}, \begin{pmatrix} 1 & \rho \\ \rho & 1 \end{pmatrix} \right)$. *For* $0 \le s < t$ *and* $-1 \le \rho \le 1$, $Pr(x \in [s,t], y \ge 0)$ *is increasing in* $\rho$, $Pr(x \in [s,t], y < 0)$ *is decreasing in* $\rho$.

*Proof.* We have

$$
\begin{aligned}
P_{s,t,+} = Pr(x \in [s,t], y \ge 0) &= \int_0^\infty \int_s^t \frac{1}{2\pi\sqrt{1-\rho^2}} e^{-\frac{x^2 - 2\rho xy + y^2}{2(1-\rho^2)}} \, dx dy \\
&= \int_0^\infty \frac{1}{2\pi\sqrt{1-\rho^2}} e^{-\frac{x^2}{2}} \int_s^t e^{-\frac{(y-\rho x)^2}{2(1-\rho^2)}} \, dy dx \\
&= \int_0^\infty \frac{1}{2\pi\sqrt{1-\rho^2}} e^{-\frac{x^2}{2}} \int_{\frac{s-\rho x}{\sqrt{1-\rho^2}}}^{\frac{t-\rho x}{\sqrt{1-\rho^2}}} e^{-\frac{u^2}{2}} \sqrt{1-\rho^2} du dx \\
&= \int_0^\infty \frac{1}{\sqrt{2\pi}} e^{-\frac{x^2}{2}} [\Phi(\frac{t-\rho x}{\sqrt{1-\rho^2}}) - \Phi(\frac{s-\rho x}{\sqrt{1-\rho^2}})].
\end{aligned}
$$

It is easy to check that this integral meets the conditions of DCT. Hence, taking the derivative yields

$$
\frac{\partial P_{s,t,+}}{\partial \rho} := \int_0^\infty \frac{1}{\sqrt{2\pi}} e^{-\frac{x^2}{2}} [\phi(\frac{t-\rho x}{\sqrt{1-\rho^2}}) \frac{-x+t\rho}{(1-\rho^2)3/2} - \phi(\frac{s-\rho x}{\sqrt{1-\rho^2}}) \frac{-x+s\rho}{(1-\rho^2)3/2}].
$$

For the first term we have

$$
\begin{aligned}
\int_0^\infty \frac{1}{\sqrt{2\pi}} e^{-\frac{x^2}{2}} \phi(\frac{t-\rho x}{\sqrt{1-\rho^2}}) \frac{-x+t\rho}{(1-\rho^2)3/2} &= \int_0^\infty \frac{1}{2\pi} e^{-\frac{(x-t\rho)^2}{2(1-\rho)^2}} e^{-\frac{t^2}{2}} \frac{-x+t\rho}{(1-\rho^2)3/2} \\
&= \frac{1}{2\pi} \frac{1}{\sqrt{1-\rho^2}} e^{-\frac{t^2}{2}} e^{-\frac{(x-t\rho)^2}{2(1-\rho^2)}} \Big|_0^\infty \\
&= -\frac{1}{2\pi} \frac{1}{\sqrt{1-\rho^2}} e^{-\frac{t^2}{2(1-\rho^2)}}. \qquad (11)
\end{aligned}
$$

Similarly we can compute

$$
\int_0^\infty \frac{1}{\sqrt{2\pi}} e^{-\frac{x^2}{2}} \phi(\frac{s-\rho x}{\sqrt{1-\rho^2}}) \frac{-x+s\rho}{(1-\rho^2)3/2} = -\frac{1}{2\pi} \frac{1}{\sqrt{1-\rho^2}} e^{-\frac{s^2}{2(1-\rho^2)}}. \qquad (12)
$$

Thus, we obtain

$$
\begin{aligned}
\frac{\partial P_{s,t,+}}{\partial \rho} &= \frac{1}{2\pi} \frac{1}{\sqrt{1-\rho^2}} e^{-\frac{s^2}{2(1-\rho^2)}} - \frac{1}{2\pi} \frac{1}{\sqrt{1-\rho^2}} e^{-\frac{t^2}{2(1-\rho^2)}} \\
&= \frac{1}{2\pi} \frac{1}{\sqrt{1-\rho^2}} (e^{-\frac{s^2}{2(1-\rho^2)}} - e^{-\frac{t^2}{2(1-\rho^2)}}) \\
&> 0,
\end{aligned}
$$

due to the fact that $s < t$. For $P_{s,t,-} := Pr(x \in [s,t], y < 0)$, we proceed with similar calculation, which will change the sign in (11) and (12) and eventually gives

$$
\frac{\partial P_{s,t,-}}{\partial \rho} = \frac{1}{2\pi} \frac{1}{\sqrt{1-\rho^2}} (e^{-\frac{t^2}{2(1-\rho^2)}} - e^{-\frac{s^2}{2(1-\rho^2)}}) < 0.
$$

The proof is complete. □

## C.2 Proof of Lemma 1

We prove a more detailed version of Lemma 1.

**Lemma C4.** *Assume $Q_{b_1}$ is a $M$-bit symmetric quantizer in the sense that it divides the positive axis into $M$ intervals with cut point $t_0 = 0 < t_1 < \cdots < t_{M-1}$. The reconstruction levels are give by $Q_{b_1}(x) = \mu_i > 0$, $x \in [t_{i-1}, t_i]$ and $Q_{b_1}(x) = -\mu_i$, $x \in [-t_i, t_{-i} - 1]$, $i = 1, ..., M$. $Q_{b_2}$ is a 1-bit quantizer such that $Q_{b_2}(y) = \nu > 0$ when $y \geq 0$ and $Q_{b_2}(y) = -\nu$ when $y < 0$. Then $E[Q_{b_1}(x)Q_{b_2}(y)]$ is strictly increasing in $\rho$ on $[-1, 1]$.*

*Proof.* Denote $P_{s,t,+} = Pr(x \in [s, t], y \geq 0)$ and $P_{s,t,-} = Pr(x \in [s, t], y < 0)$. We write explicitly

$$\mathbb{E}[Q_{b_1}(x)Q_{b_2}(y)] = \nu \sum_{i=1}^{M} \mu_i Pr_{t_{i-1},t_i,+} - \nu \sum_{i=1}^{M} \mu_i Pr_{t_{i-1},t_i,-}$$

$$- \nu \sum_{i=1}^{M} \mu_i Pr_{-t_i,-t_{i-1},+} + \nu \sum_{i=1}^{M} \mu_i Pr_{-t_i,-t_{i-1},-}$$

$$= 2\nu \sum_{i=1}^{M} \mu_i (Pr_{t_{i-1},t_i,+} - Pr_{t_{i-1},t_i,-}),$$

due to the symmetry of bivariate normal density. Since $\nu > 0$ and $\mu_i > 0$, $i = 1, ..., M$, applying Lemma C3 we prove the desired result. □

## C.3 Proof of Lemma 2

In the following we prove a detialed version of Lemma 2.

**Lemma C5.** *Consider two 2-bit symmetric quantizers $Q_{b_1}$ and $Q_{b_2}$. $Q_{b_1}$ has cut point at $(-t_1, 0, t_1)$ with distinct quantizing values $(-\mu_2, -\mu_1, \mu_1, \mu_2)$, $0 < \mu_1 < \mu_2$ on the 4 intervals separated by the cut points. Similarly, $Q_{b_2}$ has cut points $(-t_2, 0, t_2)$ and distinct codes $(-\xi_2, -\xi_1, \xi_1, \xi_2)$, $0 < \xi_1 < \xi_2$. Assume that both quantizers to be increasing, namely, $\mu_1 < \mu_2$, $\xi_1 < \xi_2$. Then $E[Q_{b_1}(x)Q_{b_2}(y)]$ is strictly increasing in $\rho$ on $[-1, 1]$.*

*Proof.* The expectations is computed as

$$\mathbb{E}[Q_{b_1}(x)Q_{b_2}(y)] = 2\mu_1\xi_1(P_{11} - p_{11}) + 2\mu_1\xi_2(P_{12} - p_{12}) + 2\mu_2\xi_1(P_{21} - p_{21}) + 2\mu_2\xi_2(P_{22} - p_{22}),$$

(13)

where

$$P_{11} = Pr(x \in [0, t_1], y \in [0, t_2]), P_{12} = Pr(x \in [0, t_1], y \in [t_2, +\infty]),$$
$$P_{21} = Pr(x \in [t_1, +\infty], y \in [0, t_2]), P_{22} = Pr(x \in [t_1, +\infty], y \in [t_2, +\infty]),$$
$$p_{11} = Pr(x \in [0, t_1], y \in [-t_2, 0]), p_{12} = Pr(x \in [0, t_1], y \in [-\infty, -t_2]),$$
$$p_{21} = Pr(x \in [t_1, +\infty], y \in [-t_2, 0]), p_{22} = Pr(x \in [t_1, +\infty], y \in [-\infty, -t_2]).$$

We compute the derivative with respect to $\rho$ for each probability using the procedure in proving lemma.

$$\frac{\partial P_{11}}{\partial \rho} = \frac{1}{2\pi} \frac{1}{\sqrt{1-\rho^2}} [e^{-\frac{t_1^2+t_2^2-2\rho t_1 t_2}{2(1-\rho^2)}} - e^{-\frac{t_1^2}{2(1-\rho^2)}} - e^{-\frac{t_2^2}{2(1-\rho^2)}} + 1]$$

$$\frac{\partial P_{12}}{\partial \rho} = \frac{1}{2\pi} \frac{1}{\sqrt{1-\rho^2}} [-e^{-\frac{t_1^2+t_2^2-2\rho t_1 t_2}{2(1-\rho^2)}} + e^{-\frac{t_2^2}{2(1-\rho^2)}}]$$

$$\frac{\partial P_{21}}{\partial \rho} = \frac{1}{2\pi} \frac{1}{\sqrt{1-\rho^2}} [-e^{-\frac{t_1^2+t_2^2-2\rho t_1 t_2}{2(1-\rho^2)}} + e^{-\frac{t_1^2}{2(1-\rho^2)}}]$$

$$\frac{\partial P_{22}}{\partial \rho} = \frac{1}{2\pi} \frac{1}{\sqrt{1-\rho^2}} e^{-\frac{t_1^2+t_2^2-2\rho t_1 t_2}{2(1-\rho^2)}}$$

$$\frac{\partial p_{11}}{\partial \rho} = \frac{1}{2\pi} \frac{1}{\sqrt{1-\rho^2}} [-e^{-\frac{t_1^2+t_2^2+2\rho t_1 t_2}{2(1-\rho^2)}} + e^{-\frac{t_1^2}{2(1-\rho^2)}} + e^{-\frac{t_2^2}{2(1-\rho^2)}} - 1]$$

$$\frac{\partial p_{12}}{\partial \rho} = \frac{1}{2\pi} \frac{1}{\sqrt{1-\rho^2}} [e^{-\frac{t_1^2+t_2^2+2\rho t_1 t_2}{2(1-\rho^2)}} - e^{-\frac{t_2^2}{2(1-\rho^2)}}]$$

$$\frac{\partial p_{21}}{\partial \rho} = \frac{1}{2\pi} \frac{1}{\sqrt{1-\rho^2}} [e^{-\frac{t_1^2+t_2^2+2\rho t_1 t_2}{2(1-\rho^2)}} - e^{-\frac{t_1^2}{2(1-\rho^2)}}]$$

$$\frac{\partial p_{22}}{\partial \rho} = -\frac{1}{2\pi} \frac{1}{\sqrt{1-\rho^2}} e^{-\frac{t_1^2+t_2^2+2\rho t_1 t_2}{2(1-\rho^2)}}.$$

Now, taking the derivative of (13) and collecting terms yields

$$\frac{\partial \mathbb{E}[Q_{b_1}(x)Q_{b_2}(y)]}{\partial \rho} = \frac{1}{\pi} \frac{1}{\sqrt{1-\rho^2}} [\mu_1 \xi_1 (A+2-2C_1-2C_2) + \mu_1 \xi_2 (2C_2-A) + \mu_2 \xi_1 (2C_1-A) + \mu_2 \xi_2 A],$$

where $A = e^{-\frac{t_1^2+t_2^2-2\rho t_1 t_2}{2(1-\rho^2)}} + e^{-\frac{t_1^2+t_2^2+2\rho t_1 t_2}{2(1-\rho^2)}}$, $C_1 = e^{-\frac{t_1^2}{2(1-\rho^2)}}$, $C_2 = e^{-\frac{t_2^2}{2(1-\rho^2)}}$. Rearranging terms, we obtain

$$\frac{\partial \mathbb{E}[Q_{b_1}(x)Q_{b_2}(y)]}{\partial \rho} \propto A(\mu_1 \xi_1 - \mu_1 \xi_2 - \mu_2 \xi_1 + \mu_2 \xi_2) + (2-2C_1-2C_2)\mu_1 \xi_1 + 2C_2 \mu_1 \xi_2 + 2C_1 \mu_2 \xi_1$$

$$= A(\mu_1 \xi_1 - \mu_1 \xi_2 - \mu_2 \xi_1 + \mu_2 \xi_2) + 2\mu_1 \xi_1 + 2C_1 \xi_1 (\mu_2 - \mu_1) + 2C_2 \mu_1 (\xi_2 - \xi_1)$$

$$> 0.$$

The last inequality holds due to $0 < \mu_1 < \mu_2$, $0 < \xi_1 < \xi_2$. $\qquad\square$

Theorem C5 requires that both quantizers be "stair-shaped" (*i.e.* increasing) functions. Next, we extend the analysis to the general case based on this result.

### C.4 Proof of Lemma 3

*Proof.* We show how to construct such decomposition. By symmetry, it suffices to consider the positive part. Suppose the cut point of $Q_b$ is $(t_1 = 0, t_2, ..., t_k)$ with values $(\mu_1, ..., \mu_k)$, all greater than 0 and in an increasing order. Now choose a number $0 < \xi_1 < \min(\mu_1, \mu_k - \mu_{k-1})$, and set the values of $Q_{b-1}$ as $(\mu_1 - \xi_1, \mu_2 - \xi_1, ...\mu_{k-1} - \xi_1)$, with cut points $(t_1 = 0, t_2, ..., t_{k-1})$. Let $Q_{b_2}$ be cut at $t_k$, with values $(\xi_1, \mu_k - \mu_{k-1} + \xi_1)$. It is easy to check that this procedure is valid in any case. This proves the lemma. $\qquad\square$

## References

[1] Theodore W. Anderson. *An Introduction to Multivariate Statistical Analysis*. John Wiley & Sons, third edition, 2003.