[Reviews · NeurIPS 2019]

Reviewer 1



This is a tricky article to review. It clearly makes some significant contributions, especially when it comes with a methodology for obtaining a quantization scheme that minimizes debiased variance. On the other hand it is very hard to read with multiple grammatical and syntactic mistakes. The main topic of the paper, asymmetric quantization, is not formally defined until page 6 and then is discussed for less than half a page. Meanwhile other results, mostly having to do with debiased estimator variance for the symmetric case, cover the lion's share of the paper. As far as I can tell these results end up having no connection with the asymmetric case. In the end this paper feels like a collection of cool results that were rushed into a paper with no common thread. In the end it doesn’t form a cohesive unit. I don’t like the idea of turning down a valuable paper for structure alone but in the end this paper just lacks the necessary polish to be published (which it ultimatly should be).

Reviewer 2



Overall an interesting paper, with useful results. I would consider the ordering question to be the most interesting contribution. However, the ordering matters when the nearest neighbors are of different classes (i.e., if the ordering of distances changes after quantization, it doesn't matter if both neighbors are in the same class). It is not clear how to properly model and analyze that, but it is worth some discussion in the paper. ==== I've seen and taken into account the author's response, and it does not change my score.

Reviewer 3



I think this paper is in good quality. The problem is not very complicated, but the authors studied the problem thoroughly, from theoretic analysis of mean and variance in different scenarios, to proposed refined method and experimental studies. I think this paper meets the standard of NeurIPS. Here are some minor suggestions. 1. I wish to see more discussions about previous work on quantized random projections, especially for cosine-similarity or inner-product estimation. The authors provide some references in the introduction, but I wish to see more detailed comparisons of results of the previous work and the current work. This will make the main contribution of this work more clear. 2. In Figure 1, please make the y-axis of the first two figures aligned. Also for Figure 2. 3. In the supplemental materials, the section numbers are wrong. (should be Section 3 in the title of Section A, etc.) ============================================================== I'm satisfied with the authors' feedback and I would like to increase my score from 6 to 7. I vote to accept this paper because of its high technical quality. I wish the authors could improve the paper's organization and presentation to match the technical quality of this paper.

[Author Response · NeurIPS 2019]

# 1   Response to Reviewer 1

Thank you very much for your comments and suggestions. From the review, one main concern is that *"main topic of the paper, asymmetric quantization, is not formally defined until page 6 and then is discussed for less than half a page"*.

We feel this might be a misunderstanding. This paper is about "Asymmetric Quantization" which include two scenarios. On Page 1, the paper says *"we consider recovering inner products from asymmetric quantized random projections (a detailed introduction of quantization is given in next section) in following circumstances"*:

- **Scenario 1: quantization vs full-precision**. We provide examples and references for "Scenario 1" on Page 1. Then the entire Section 3 (Pages 3-6) is devoted to the analysis and results for Scenario 1.
- **Scenario 2: quantization with different bits**. We provide examples and references on Page 2 and allocate Section 4 (Page 6) for Scenario 2. It appears Section 4 is what the Reviewer considered as "asymmetric quantization". If so, this is the major misunderstanding as asymmetric quantization includes both scenarios.

Thus, we feel the misunderstanding may be that the Reviewer did not consider Scenario 1 (quantization vs full-precision) as asymmetric quantization. But indeed Scenario 1 is a very important special case with many applications. From the analysis perspective, Scenario 1 is, to a large extent, easier to analyze than Scenario 2. In fact, some of the analysis and results (Section 3) derived for Scenario 1 are subsequently used by some of the analysis (Section 4) in Scenario 2.

The other main concern from the review says *"Meanwhile other results, mostly having to do with debiased estimator variance for the symmetric case, cover the lion's share of the paper. As far as I can tell these results end up having no connection with the asymmetric case."* This appears to be another misunderstanding. Please allow us explain.

Essentially the entire paper is about "asymmetric quantization". In the paper, we focused on the analysis of the "debiased estimator variance" for Scenario 1: quantization vs. full-precision (infinite number of bits). It is fair to say it *"covers the lion's share of the paper"*, but it is for asymmetric quantization. We hope this clarifies your concerns.

Therefore, for your question *"Is the paper about debiased variance, asymmetric quantization, monotonicity of inner product estimates?"*, the answer is that this paper is about asymmetric quantization with the special important scenario of "quantization vs. full-prevision" which benefits substantially from the analysis of "debiased variance". Unlike the symmetric case, the monotonicity result under asymmetric quantization requires a nontrivial proof. Only after we have proved the monotonicity of inner product estimators, practitioners can be assured when using asymmetric quantization.

To address *"Minor suggestions"*. Firstly, thanks so much for suggestion on notation, figure caption, etc. As for mis-ordering probability, it is doable to consider $n > 2$ data points, which would look much more complicated as a sum of probabilities. It is sufficient to present result for two points in order to show connection with debiased variance— if the mis-ordering probability between $(x, y)$ is higher for $\forall y \neq x$ when $x$ is the true nearest neighbor, then the chance of estimating a wrong neighbor would trivially be higher. Hence, in this paper we present the 2-points case for conciseness.

Thanks again for raising the main concerns. We hope it is now clear that the paper is indeed all about asymmetric quantization, with "quantization vs. full-precision" being an important special case (which covers a bulk of the paper).

# 2   Response to Reviewer 2

We appreciate your valuable comments and suggestions. The two research problems you mentioned: (a) the impact of mis-ordering on mis-classification error, (b) the trade-off between $k$ and $b$ at a given budget of $k \times b$, are both good questions whch are worth discussions. For question (b), since we have essentially derived the theoretical variances with respect to $k$ and $b$, in principle we can provide additional plots to illustrate the the trade-off. For question (a), in this paper we focus on similarity search for retrieval tasks, which have no class label information. To extend the analysis to classification problems require additional efforts which might be well-suited for future work. Thank you.

# 3   Response to Reviewer 3

Thanks very much for a precise summary of our work and various nice suggestions for improving the paper. In this submission, we cited a total 39 papers which are related to our work. Despite the page limit, we will follow your suggestion by trying harder to further expand the discussions on related works. Particularly thanks for your kind suggestion that we could align the y-axis in Figure 1 and Figure 2. It is a good suggestion. In the submission, we tried to maximally make use of the space available in the figures to display the curves, but you are right that it might be better to align the y-axis to provide a more direct overview of the two plots. Thank you.

[Meta-Review · NeurIPS 2019]

This is a strong theoretical paper with important theoretical contributions. One reviewer (Reviewer 1) had some reservations which were mainly due to the structure of the paper and the quality of writing. After the rebuttal and the discussion, that reviewer was expressing a weak accept position given the importance of the results presented. I suggest that the paper is accepted as a poster. [This meta-review was reviewed and revised by the Program Chairs]